# Intermolecular cascaded $\pi$-conjugation channels for electron delivery powering $CO_2$ photoreduction

Shengyao Wang [1,2], Xiao Hai [2,3], Xing Ding[1], Shangbin Jin[4], Yonggang Xiang[1], Pei Wang[1], Bo Jiang[2], Fumihiko Ichihara[2,3], Mitsutake Oshikiri[2,5], Xianguang Meng[2], Yunxiang Li[2,3], Wakana Matsuda[6], Jun Ma[6], Shu Seki [6], Xuepeng Wang[1], Hao Huang[2], Yoshiki Wada[7], Hao Chen[1✉] & Jinhua Ye [2,3,8✉]

Photoreduction of $CO_2$ to fuels offers a promising strategy for managing the global carbon balance using renewable solar energy. But the decisive process of oriented photogenerated electron delivery presents a considerable challenge. Here, we report the construction of intermolecular cascaded $\pi$-conjugation channels for powering $CO_2$ photoreduction by modifying both intramolecular and intermolecular conjugation of conjugated polymers (CPs). This coordination of dual conjugation is firstly proved by theoretical calculations and transient spectroscopies, showcasing alkynyl-removed CPs blocking the delocalization of electrons and in turn delivering the localized electrons through the intermolecular cascaded channels to active sites. Therefore, the optimized CPs (N-CP-D) exhibiting CO evolution activity of 2247 $\mu$mol g$^{-1}$ h$^{-1}$ and revealing a remarkable enhancement of 138-times compared to unmodified CPs (N-CP-A).

[1] College of Science, Huazhong Agricultural University, Wuhan 430070, P. R. China. [2] International Center for Materials Nanoarchitectonics (WPI-MANA), National Institute for Materials Science (NIMS), 1-1 Namiki, Tsukuba, Ibaraki 305-0044, Japan. [3] Graduate School of Chemical Sciences and Engineering, Hokkaido University, Sapporo 060-0814, Japan. [4] Key Laboratory of Material Chemistry for Energy Conversion and Storage, Ministry of Education, School of Chemistry and Chemical Engineering, Huazhong University of Science and Technology, Wuhan 430074, P. R. China. [5] International Center for Material Nanoarchitectnoics (WPI-MANA), National Institute for Materials Science (NIMS), 3-13 Sakura, Tsukuba, Ibaraki 305-0003, Japan. [6] Department of Molecular Engineering, Kyoto University, Kyoto 615-8510, Japan. [7] Electroceramics Group, National Institute for Materials Science (NIMS), 1-1 Namiki, Tsukuba, Ibaraki 305-0044, Japan. [8] TJU-NIMS International Collaboration Laboratory, School of Materials Science and Engineering, Tianjin University, Tianjin 300072, P. R. China. ✉email: hchenhao@mail.hzau.edu.cn; jinhua.ye@nims.go.jp

Global carbon dioxide ($CO_2$) emissions from burning fossil fuels reached 33 gigatons in 2017, twice the natural rate at which $CO_2$ is adsorbed back into land and ocean sinks. Harnessing solar radiation holds the answer to reducing our dependence on fossil fuels and reducing greenhouse gas emissions. The utilization of photoexcited high-energy photoelectrons with the help of semiconductors to drive the energy conversion of $CO_2$ to value-added green fuels is believed to be an effective strategy for promoting sustainable development in society[1–5]. Beyond conventional semiconductors, conjugated polymers (CPs), as a new class of organic semiconductors, is expected to be the next generation of multifunctional photocatalyst because of their versatile photophysical properties, intriguing electronic properties, and especially the adjustable monomer structure, endowing it with manageable light absorption and controllable electronic localization ability. Therefore, numerous studies on developing various strategies for capable of photocatalytic energy conversion over CPs have been carried out[6–8]. Since Cooper's group first found a series of pyrene-based CPs active in hydrogen ($H_2$) evolution under visible light irradiation with platinum as cocatalyst[9], researchers have tried to anticipate to the application of such materials in $CO_2$ photoreduction[10]. For CPs in photocatalytic $CO_2$ reduction, the light conversion efficiency is rate-determined by the photoexcited electrons delivery from CPs to the surface loaded cocatalyst (adsorb and active $CO_2$ molecule). However, a limitation still exists in finding an appropriate way to promote the delivery of photoexcited electrons to cocatalyst due to a higher energy barrier of the out-of-plane Ohm or Schottky contact than the intramolecular cascade between cocatalyst and CPs[11,12].

Photoinduced intermolecular charge transfer through molecules by non-covalent interactions is a well-known efficient process in photochemistry[13]. To achieve kinetically favorable electron transfer from CPs to cocatalyst and make a breakthrough in $CO_2$ photoreduction, an intermolecular cascaded channel between the CPs and cocatalyst is desirable to be established for oriented delivery of photoexcited electrons to overcome a lower energy barrier and a less carrier[14–17]. Transition metal bipyridine compounds with a π-conjugated structure, such as Co (II) bipyridine complexes have been recognized as one of the most active centers for adsorbing and activating $CO_2$ molecular and even achieving photocatalytic $CO_2$ reduction in the presence of some photosensitizer[18–21]. However, the photocatalytic $CO_2$ reduction process is severely hampered due to the instability of the light-absorbing material and the obstruction of interface electron transport[17]. When analyzing the spatial structure of π-conjugated Co (II) bipyridine complexes cocatalyst appeared in the large π-conjugated pyrene-based CPs, the most striking feature is that the strong π–π interactions will self-assemble them into an intermolecular π–π stacking structure[22–25]. Inspired by this, we speculated that an electronic transmission channel could be built via the enhanced π-electronic cloud interactions to ensure the photoexcited electrons freely deliver from CPs to cocatalysts[26]. Furthermore, for CPs, the absence of unsaturated bond between two adjacent aromatic rings can not only reduce the steric repulsion (reducing the twist angle of adjacent aromatic rings) but

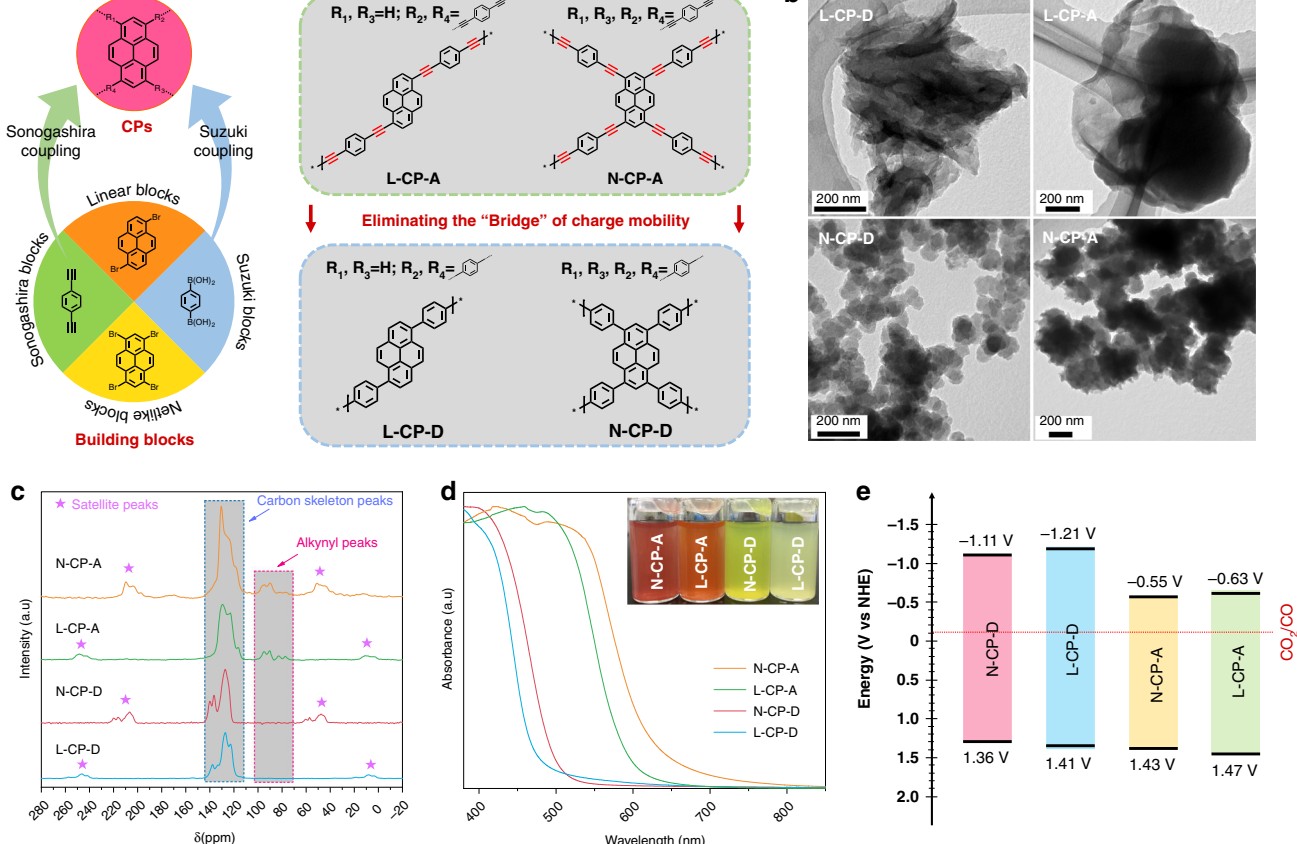

**Fig. 1 Preparation and characterization of the CPs. a** Illustration of synthesis and the strategy of eliminating the charge-transfer bridge. **b** TEM images of CPs. **c** Solid-state $^{13}$C corss-polarization/magic angel spinning nuclear magnetic resonance (CP/MAS NMR) spectroscopy of CPs. **d** Solid-state UV/Visible diffuse reflectance spectra (DRS) of CPs, Inset: CPs dispersed in an acetonitrile/water (7:3) mixture. **e** Highest occupied molecular orbital (HOMO) and lowest unoccupied molecular orbital (LUMO) band position diagram for CPs obtained from cyclic voltammetry (CV) and DRS.

also weaken the intramolecular conjugate interaction[27,28]. The weakened conjugate interaction between adjacent aromatic rings of CPs is expected to result in free-π-electrons localized that could improve its intermolecular cascading ability with Co (II) bipyridine complexes[29].

To validate the above strategy, four goal-oriented materials including linear and net-like (Net-like materials are usually referred to as conjugated microporous polymers) CPs with simple structure[24,25], but different π-conjugation are built by using Suzuki-Miyaura coupling instead of Sonogashira-Hagihara coupling in synthesis. From first-principles theoretical calculation and experimental data, we prove that the CPs without alkynyl groups strictly block the delocalization of photoexcited electrons due to the lack of intramolecular charge-transfer bridges, which in turn deliver the photoexcited electrons faster to Co (II) bipyridine complexes through the intermolecular cascaded channels, leading to a state-of-the-art $CO_2$ photoreduction activity. This strategy constructs an efficient system of $CO_2$ photoreduction over Co (II) bipyridine complexes and pyrene-based CPs with modification of both intramolecular and intermolecular conjugations. Our results also provide evidence and mechanism of enhanced charge transfer via the pathway of non-covalent interactions. The built-in intermolecular cascaded channels worked out the most critical challenge in the electron delivery from CPs to cocatalyst, providing a point of view in the construction of CPs-based system for $CO_2$ photoreduction.

## Results

### Synthesis, structure, and spectroscopic properties of CPs.

The four kinds of pyrene-based CPs with or without alkynyl groups were designed based on classic Sonogashira-Hagihara and Suzuki-Miyaura coupling processes, and linear or net-like structures were modified by changing the building blocks (see the "Methods" section for experimental details), as outlined in Fig. 1a[9,30]. Linear (L-CP-A) and net-like (N-CP-A) alkynyl-connected CPs can be obtained by using the Sonogashira blocks (1,4-diethynylbenzene) polymerized with the linear blocks (1,6-dibromopyrene) and net-like blocks (1,3,6,8-tetrabromopyrene), respectively (Supplementary Fig. 1). As a comparison, Suzuki blocks (1,4-phenylenediboronic acid) were employed as substitutes for Sonogashira blocks and successfully construct the corresponding linear (L-CP-D) and net-like (N-CP-D) directly connected CPs without alkynyl groups.

The transmission electron microscopy (TEM) and scanning electron microscopy (SEM) revealed that the CP-A series (L-CP-A and N-CP-A) exhibited a more agglomerated state than did the CP-D series (D-CP-A and D-CP-A) because of the higher π conjugation. (Fig. 1b and Supplementary Fig. 2). Although the CPs presented different agglomeration states at low resolution, the high-resolution transmission electron microscopy (HR-TEM) and the powder X-ray diffraction profile (PXRD) revealed that each CPs exhibits the basic characteristics of amorphous carbon, which means similar structure of CPs were constructed during the synthesis process (Supplementary Figs. 3 and 4)[31]. For a more in-depth comparison of the structural differences in CPs, we utilized solid-state $^{13}C$ cross-polarization/magic angle spinning nuclear magnetic resonance ($^{13}C$ CP/MAS NMR) spectroscopy to demonstrate the exact structure (Fig. 1c)[32]. Referencing the estimated chemical shifts of different carbons in CPs (Supplementary Fig. 5), the similar chemical shifts of all these CPs between 110 and 130 parts per million (ppm) can be assigned to the aromatic carbons of the phenyl and pyrenyl units[33]. The peaks at ~140 ppm, which only existed in the CP-D series, were ascribed to the mutually substituted aromatic carbon. For the CP-A series, there are some signals at ~90 ppm can be indexed to the

characteristic peak of alkynyl[34]. Notably, the locations of the satellite peaks attribute to spinning sidebands in N-CP-A were consistent with those in N-CP-D. A similar phenomenon was also found for L-CP-A and L-CP-D, which may be attributed to the structural difference between linear and net-liked CPs. Besides, the CPs with or without alkynyl was also confirmed by Raman spectroscopy, X-ray photoelectron spectroscopy (XPS) and Fourier transform infrared (FT-IR) spectroscopy (Supplementary Figs. 6–8)[35,36].

After fully identifying the structure of CPs, we employed UV–Visible diffuse reflectance spectra (DRS) to monitor the light absorption of CPs[37]. As shown in Fig. 1d, the absorption band edges of CPs were located in the visible region ranging from 470 to 620 nm, which was consistent with the color of the CPs (inset of Fig. 1d). According to the Kubelka-Munk equation, the absorption band edges of L-CP-A and N-CP-A correspond to bandgaps of 2.10 and 1.98 eV, respectively. While the bandgaps of L-CP-D and N-CP-D were estimated to be 2.62 and 2.47 eV, respectively, which is larger than CP-A series due to the decreased conjugation (Supplementary Fig. 9)[38]. Cyclic voltammetry (CV) measurements were also conducted, the highest occupied molecular orbital (HOMO) position can be determined by the irreversibility of the oxidation peaks because the irreversible oxidation process of the CPs at the impressed voltage (Supplementary Fig. 10) revealed different energy levels within the CPs (Supplementary Table 1)[39]. In addition, their energy levels were further investigated by the ultraviolet photoelectron spectroscopy (UPS) (Supplementary Fig. 11) and Mott-Schottky test (Supplementary Fig. 12), which showed a high accordance with the energy levels determined by CV measurement (Supplementary Table 2). Although the CP-D series exhibited higher lowest unoccupied molecular orbital (LUMO) levels than the CP-A series, all of these CPs had enough negative potentials to carry out the reduction of $CO_2$ to CO (Fig. 1e).

### Charge mobility and intermolecular cascaded channel of CPs.

Based on the conjugation of CPs, we can speculate the electron localization of CPs could be increased by eliminating the alkynyl group. For further confirmation of the weakened conjugate interaction and free-π-electrons localization of these CPs without the alkynyl group, the electrodeless flash-photolysis time-resolved microwave conductivity (FP-TRMC) was employed to evaluate photoexcited electrons transport in CPs. Unlike the conventional techniques that are highly affected by the influence of factors such as impurities, chemical or physical defects, and organic/electrode interfaces, FP-TRMC allows for probing the motion of the charge carrier before complete deactivation by trapping sites[40,41]. The conductivity transients and calculated charge mobilities for CPs are displayed in Fig. 2a, in which the L-CP-A, with a linear structure and alkynyl group, exhibits charge mobility ($\Sigma\mu$) of 0.32 $cm^2 V^{-1} s^{-1}$ ($\phi\Sigma\mu = 7.4 \times 10^{-5} cm^2 V^{-1} s^{-1}$). As expected, the charge mobility of L-CP-D in the absence of alkynyl decreased to a much lower value of 0.15 $cm^2 V^{-1} s^{-1}$ ($\phi\Sigma\mu = 3.4 \times 10^{-5} cm^2 V^{-1} s^{-1}$). For N-CP-A CPs, the network structure provided two additional pathways than that of linear structured L-CP-A for electronic delocalization due to the two more connections of the alkynyl group to each pyrenyl units, thus resulting in the maximum value of 0.35 $cm^2 V^{-1} s^{-1}$ ($\phi\Sigma\mu = 7.6 \times 10^{-5} cm^2 V^{-1} s^{-1}$) among all CPs. In comparison, the N-CP-D possesses relative lower charge mobility of 0.25 $cm^2 V^{-1} s^{-1}$ ($\phi\Sigma\mu = 5.5 \times 10^{-5} cm^2 V^{-1} s^{-1}$) as a result of the weak intramolecular conjugate interaction. Moreover, the CPs were subjected to additional electrochemical analyses, such as photocurrent measurement and the electrochemical impedance spectroscopy (EIS) Nyquist plots, which

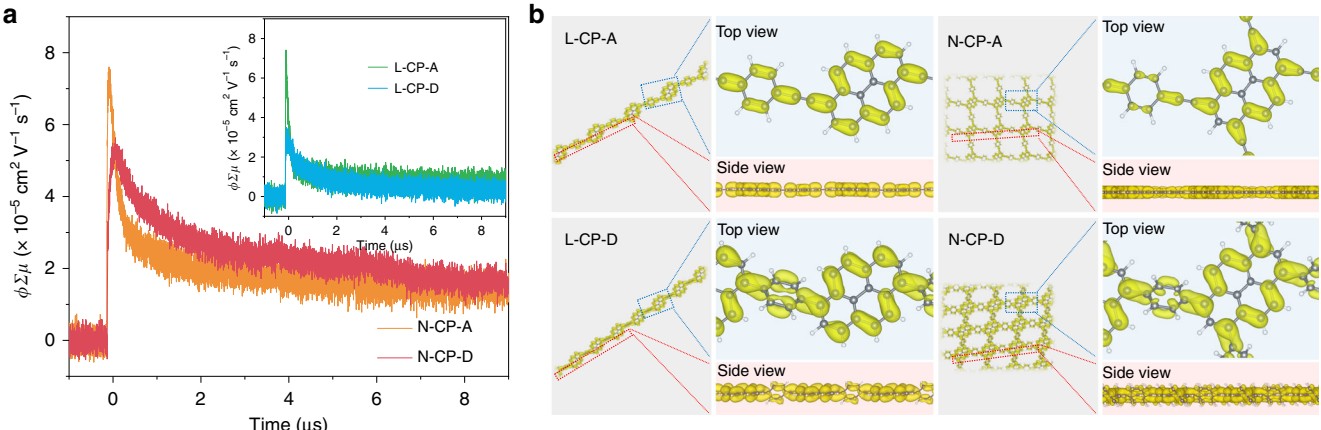

**Fig. 2 Characterization of electronic delocalization in CPs. a** Conductivity transients observed by flash-photolysis time-resolved microwave conductivity (FP-TRMC) spectroscopy upon excitation at 355 nm laser pulses at $1.8 \times 10^{16}$ photons cm$^{-2}$ for CPs. **b** Top and side views of the charge distribution of LUMO of CPs at the Γ k-point.

further validated that, consistent with the FP-TRMC results, the elimination of the alkynyl group in CPs greatly enhanced localization of free-π-electrons (Supplementary Figs. 13 and 14).

To gain insight into the effect of the alkynyl group on the localization of free-π-electrons, first-principles theoretical calculations based on hybrid functional were subsequently performed to compare the electronic localization of the CPs (see the "Methods" section for experimental details)[35]. As shown, the lowest unoccupied molecular orbital (LUMO) and highest occupied molecular orbital (HOMO) changes (Fig. 2b and Supplementary Fig. 15) at the Γ k-point distribution unambiguously demonstrated the apparent difference in the localization of free-π-electrons in CP-D series and CP-A series, respectively. From the top view of the charge distribution profiles of the two linear CPs (L-CP-A and L-CP-D), both LUMO and HOMO of L-CP-D exhibited stronger electron distribution asymmetry than those of L-CP-A as the conjugation changed. This means that L-CP-D could concentrate more electrons in a particular section under the light irradiation. Moreover, a similar phenomenon can also be observed from the comparison of two net-like CPs (N-CP-A and N-CP-D). In the light of the above results and electronic property comparison of backbone architecture with or without alkynyl (Supplementary Fig. 16), the energy gap between unoccupied and occupied orbital at the edge and corner of the pyrenyl is smaller than that at central area, the pyrenyl could play a role of a kind of antenna to collect the excited carriers at around the edge and corner. This is an advantage to achieve charge separation and harvest a wide range of photon energy.

Though the comparative analysis of computed interlayer interaction energies, it was consistency with what we expected (Supplementary Fig. 17 and Table 3). The existence of alkyne in CP-A series favors the electrons transfer in intramolecular. However, the CP-D series without alkynyl as a connector could possess more local photoelectrons, thus the interlayer interaction energy of bilayer CP-D series was much strong than that of CP-A series. This might indicate electron transfer along intermolecular of bilayer CP-D series more easily than that of bilayer CP-A series. Thus, we could propose the following situation on different CPs. For the CP-A series, photoelectrons were generated on the central light-responsive part of pyrenyl and then transferred to the other parts with alkynyl as a bridge, which results homogeneously charge density of CP-A series. In contrast, the absence of alkynyl in CP-D series led to a retardation of intramolecular electron delocalization. Moreover, it also gave rise to the increased charge density in some parts of CP-D series

under light irradiation, which is favorable for improving the electronic delivery over the built-in intermolecular cascaded channels via π–π interactions between CPs and cocatalyst.

**CO$_2$ Photoreduction activity over CPs**. To study if the electron delivery from the CPs to cocatalyst has critical effects on CO$_2$ photoreduction properties, the evaluation of CO$_2$ photoreduction activities (see the "Methods" section for experimental details) were carried out in a closed gas circulation system by using CPs as the catalyst and 5 μmol Co (II) bipyridine complexes as cocatalyst. The acetonitrile/water (7:3) mixture with triethanolamine (TEOA) as sacrificial agent were also added. (Supplementary Fig. 18)[42,43]. The Co (I) bipyridine complexes produced by reduction of Co (II) bipyridine complexes are very powerful reducing agents which could be excellent candidates cocatalyst for photoreduction reaction based on the previous report[43]. We also demonstrated that photo-excited electrons on the LUMO of CPs in the present work do have the ability to reduce Co (II) bipyridine complexes to Co (I) bipyridine complexes via the cyclic-voltammetry spectrum (Supplementary Fig. 19). In addition, the weakened EPR signal (Supplementary Fig. 20, see the "Methods" section for experimental details)[22] indicates the weakening of high-spin-state Co (II) upon visible-light irradiation that maybe reduced to low-spin-state Co (I). In order to get more convincing evidence for the existence of the low-spin-state of Co (I), the in-situ XPS (Supplementary Fig. 21) were performed over N-CP-D. It showed a lower binding energy than that of Co$^{2+}$, which clearly revealed the existence of photo-induced low-spin-state Co$^+$ in this system. In some recent researches, the Co (II) bipyridine complexes were selected as the model cocatalyst to study the CO$_2$ photoreduction activity of the materials[17,42]. Therefore, we speculate that after optimizing the π-conjugation of CPs, the system consist of CPs and Co (II) bipyridine complexes will exhibit good potential for CO$_2$ photoreduction. As displayed in Fig. 3a, after 5 h of visible light (>420 nm) irradiation, the N-CP-D generated a maximum amount of carbon monoxide (CO) of 56.8 μmol (11.37 μmol h$^{-1}$), while the L-CP-D generated 20.2 μmol (4.03 μmol h$^{-1}$) of CO. In contrast, only 0.3 and 0.4 μmol of CO was detected when L-CP-A and N-CP-A were used as the catalyst, respectively. The CO evolution rates of N-CP-D and L-CP-D were almost 138 times and 81 times higher than those of N-CP-A and L-CP-A, respectively. These CO$_2$ photoreduction rates and enhancements are comparable to that of many similar systems containing photocatalyst and cobalt-complexes

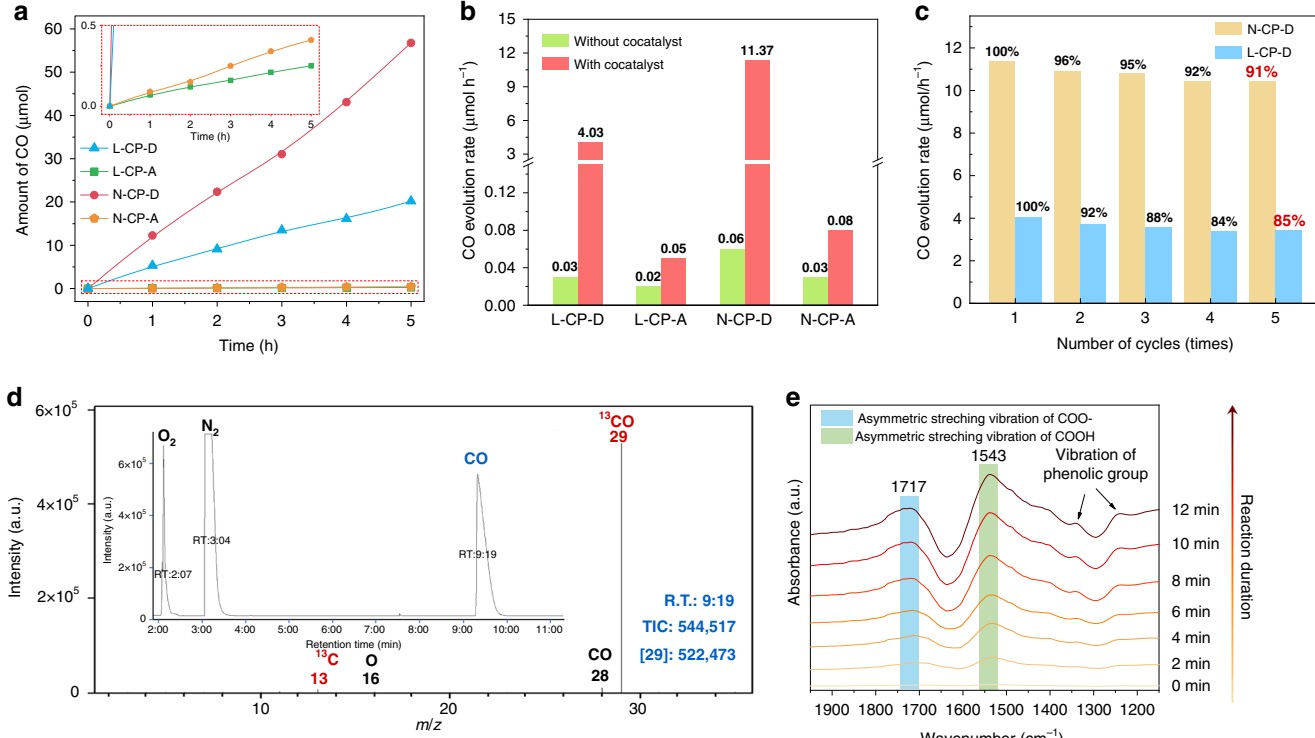

**Fig. 3 CO₂ photoreduction performance of the CPs. a** Time course of the produced CO for CPs during 5 h experiment performed under visible light (420 nm cut-off filter) in an acetonitrile/water (7:3) mixture using triethanolamine (TEOA) as sacrificial agent and 5 µmol Co (II) bipyridine complexes as a cocatalyst over 5 mg CPs under ~80 KPa of pure $CO_2$ gas, inset: X-axis enlarged performance of CO evolution. **b** Average CO evolution rates for CPs with (red bar) or without (green bar) cocatalyst, **c** Recyclability of N-CPD (yellow bar) and L-CP-D (blue bar) in the photocatalytic $CO_2$ reduction within five cycles. **d** Mass spectra of $^{13}CO$ and total ion chromatography (inset) over N-CP-D in the photocatalytic reduction of $^{13}CO_2$. **e** In-situ diffuse reflectance infrared Fourier transform spectroscopy (DRIFTS) for the photocatalytic reduction of $CO_2$ over N-CP-D.

cocatalyst, especially π-conjugated Co (II) bipyridine complexes. It can be found that, although the addition of Co (II) bipyridine complexes significantly improved the photocatalytic $CO_2$ reduction activity of the original photocatalyst, the photocatalytic $CO_2$ reduction activity severely hampered by the obstruction of interface electron transport caused a relative lower multiple of increases (Supplementary Table 4). Owing to the solution of the oriented electron delivery, the activity of $CO_2$ reduction reaction over the CPs with optimized conjugation showed a most considerable enhancement.

To further uncover the underlying reasons for the substantial difference that we obtained between the CP-D series and the CP-A series, we applied CPs in the $CO_2$ photoreduction without a cocatalyst (Fig. 3b). All CPs showed quite low activity in the absence of cocatalyst, as well as the CP-D series in the presence of isolated cobalt chloride or dipyridyl (Supplementary Fig. 22), implying that the residual Pd has negligible effect on the photocatalytic $CO_2$ reduction activity (Supplementary Table 5). After the addition of Co (II) bipyridine complexes as a cocatalyst, although all of CPs exhibit enough negative potential to transfer the photoelectrons to cocatalyst, only an ~2.5-fold increase was found for L-CP-A and N-CP-A. However, for L-CP-D and N-CP-D, the CO evolution rate increased by 134 times and 190 times, respectively. With the addition of appropriate cocatalyst (Supplementary Table 6), the CO selectivities of L-CP-D and N-CP-D were measured to be 86% and 82% (Supplementary Fig. 23) and achieved an apparent quantum yield (AQY) as high as 3.39% and 1.23% at 400 nm, respectively (Supplementary Fig. 24), which is considered higher than that in most reports until now (Supplementary Table 4). Moreover, the stability test over L-CP-D and N-CP-D indicates that after 5 cycles, the CO evolution

was still high (maintaining 91% to the original cycle without adding fresh Co complexes to the system) as compared to the initial values (Fig. 3c), suggesting the adequate stability of these CPs (Supplementary Fig. 25). Therefore, we speculate that upon eliminating the alkynyl group in the structure of CPs, the weakened intramolecular conjugation of CP-D series blocks the delocalization of photoexcited electron. Stacked electrons on CP-D series were not well dispersed and thus tended to be fast delivered to cocatalyst through the intermolecular cascaded channels, making the CP-D series performed far better activity than did the CP-A series (Supplementary Fig. 26). Besides, the net-like CPs (N-CP-A and N-CP-D) possess more cocatalyst absorption sites of phenyl in the units, thus resulting in enhanced performance of net-like CPs (N-CP-A and N-CP-D) than those of linear CPs (L-CP-A and L-CP-D).

To validate the generation of CO originated from the catalytic splitting of $CO_2$, we employed isotope-labeled carbon dioxide ($^{13}CO_2$) as a substitute source gas with the N-CP-D (Fig. 3d) or L-CP-D (Supplementary Fig. 27) to complete the evaluation experiment (see the "Methods" section for experimental details)[44]. As shown in Fig. 3d, the total ion chromatographic peak around 9.3 min can be assigned to the CO (inset of Fig. 3d). Moreover, the main peak at $m/z = 29$ achieved a high abundance, and the fragments produced from this peak ($^{13}C$ at $m/z = 29$ and O at $m/z = 16$) in the mass spectra indicated that the produced $^{13}CO$ indeed originated from $^{13}CO_2$ in the $CO_2$ photoreduction over N-CP-D. A similar phenomenon can also be observed in using L-CP-D as catalyst and intensity of $^{13}CO$ is one-third of using N-CP-D (Supplementary Fig. 27), which is highly consistent with the above activity measurement. Subsequently, we also used the in-situ diffuse reflectance infrared Fourier transform spectroscopy (DRIFTS) to

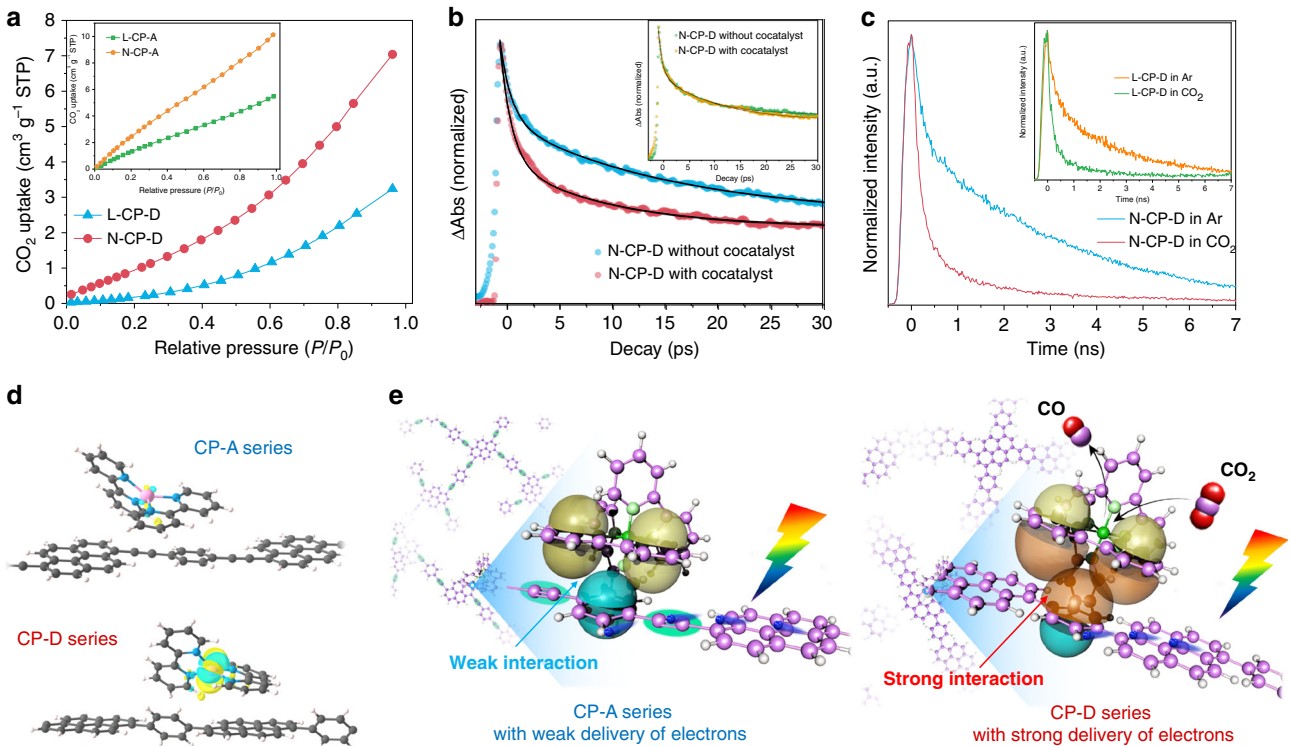

**Fig. 4 Electron delivery from CPs to cocatalyst for CO2 photoreduction. a** $CO_2$ adsorption capacities of the CP-D series and CP-A series (inset) at 273.15 K. **b** Kinetics of electrons in transient absorption over N-CP-D with or without cocatalyst under a probe wavelength of 2200 nm, inset: Kinetics of electrons in transient absorption over N-CP-A with or without cocatalyst under a probe wavelength of 2500 nm. **c** Time-resolved photoluminescence decay of L-CP-D and N-CP-D (inset) under a $CO_2$ or argon atmosphere. **d** Charge density difference of CPs in present and absent of Co (II) bipyridine complexes with the isosurfaces value of 0.001 e/Å³. yellow and cyan represent charge accumulation and charge depletion, respectively. The pink, indigo, dark gray and white pink balls represent Co, N, C and H atoms, respectively. **e** Proposed process of electron transfer over the CP-A series and CP-D series for the $CO_2$ photoreduction reaction.

unveil the process of $CO_2$ photoreduction occurring on N-CP-D with a cocatalyst (see the "Methods" section for experimental details). Based on the spectra collected for the N-CP-D with or without absorbed CO, the in situ generated CO increased with the illumination time (Supplementary Fig. 28) and the main intermediates of this process were assigned to the asymmetric stretching vibrations of $COO^-$ and COOH species in Fig. 3e[45], which is solidly in accordance with the mechanism of $CO_2$ conversion to CO in the previous report[46,47].

**Electron delivery from CPs to cocatalyst for $CO_2$ reduction.** By discovering and scrutinizing the $CO_2$ photoreduction of CPs, we hope to verify our conjecture that the localized free-π-electrons of CPs could improve its intermolecular cascading ability with cocatalyst for $CO_2$ photoreduction. The adsorption of $CO_2$ is a prerequisite for $CO_2$ photoreduction. Although the CP-D series have a larger specific surface area (Supplementary Fig. 29 and Table 7) due to a shorter skeleton length[48], the CP-A series exhibited greater $CO_2$ adsorption than the CPs-D series (Fig. 4a), attributing to the high conjugation from bifunctionalization with pyrenyl and alkynyl group[49]. Nevertheless, the $CO_2$ absorption of a dry powder is very different from the wet conditions[50], the in-situ FT-IR indicates L-CP-D and N-CP-D exhibit enhancement both in $CO_2$ adsorption and $CO_2$ chemisorption than L-CP-A and N-CP-A (Supplementary Fig. 30). It means that the $CO_2$ adsorption capacity of CPs-D series is stronger than that of CPs-A series under the solvent-containing environment, which provides favorable conditions for the subsequent $CO_2$ reduction reaction. Femtosecond transient absorption (TA) spectroscopy is

a useful technique for studying ultrafast charge transfer in interfaces. We thus employed this method (see the "Methods" section for experimental details) to verify the different kinetics in intermolecular electron delivery for the solid evidence of built-in intermolecular cascade channel. From the TA spectra dependence on the wavelength (Supplementary Fig. 31), the main state of trapped electrons can be assigned to the wavelength of 2500 and 2200 nm, while the corresponding state of holes at 550 and 525 nm for N-CP-A and N-CP-D, respectively. Considering these results, we measured the kinetics of electrons over N-CP-D with or without cocatalyst under a probe wavelength of 2200 nm. In Fig. 4b, compared to the initial one, the decay of N-CP-D with cocatalyst showed a noticeable decrease, while N-CP-A showed no difference with or without cocatalyst (inset of Fig. 4b). Combining the results for the corresponding holes (Supplementary Figs. 32 and 33), the enhanced electron delivery over N-CP-D was determined. Moreover, we employed time-resolved photoluminescence (TR-PL) spectroscopy to gain insight into the subsequent process of electrons transfer from the cocatalyst to $CO_2$ (see the "Methods" section for experimental details)[51]. For both N-CP-D (Fig. 4c) and L-CP-D (inset of Fig. 4c), the average lifetime of electrons (Supplementary Table 8) in a $CO_2$ atmosphere was significantly shortened compared to that in an argon atmosphere, which can be mainly attributed to the large amount of photogenerated electrons can be fast delivered to $CO_2$ via the cocatalyst and only a small number of electrons with short lifetime involved into the detectable recombination process under the $CO_2$ atmosphere (Supplementary Fig. 34), which can be further confirmed by the TA spectra in different atmosphere (Supplementary Fig. 35). However, both N-CP-A and L-CP-A in

TR-PL measurements showed no difference between the atmosphere of $CO_2$ and argon due to the weak delivery (Supplementary Fig. 36).

In order to gain a deeper insight into the built-in intermolecular cascaded channels between CPs and Co (II) bipyridine complexes, the charge distribution analysis and charge density difference (Fig. 4d and Supplementary Figs. 37 and 38) of before and after adsorption of Co (II) bipyridine complexes over L-CP-D (0.12 e) and L-CP-A (0.07e) were used to figure out that more charges were delivered from the CP-D series to the adsorbed Co (II) bipyridine complexes than that of CP-A series. It indicates that the CP-A series in presence of alkynyl facilitate its intramolecular charge mobility, while the CP-D series in absence of alkynyl promote its intermolecular cascading ability with the Co (II) bipyridine complexes. Therefore, we can rationally propose that the $CO_2$ photoreduction over CPs was promoted by the built-in intermolecular cascaded channels, as shown in Fig. 4e. When using the CP-A series, the photoelectrons were generated and quickly transferred to the other part of the CPs in the presence of alkynyl groups as the bridge. Since the electrons were distributed throughout the CPs, it was difficult to be delivered to the cocatalyst through the weak intermolecular π–π interactions in CP-A series. In contrast, by eliminating the alkynyl group, the CP-D series could prevent photoexcited electrons from transferring to other parts of CPs due to lack of intramolecular charge-transfer bridges. An intermolecular cascaded channel could be built via the enhanced π-electronic cloud interactions between CPs and cocatalyst to ensure the delivery of photoexcited electrons. As a result, a cascaded electron supply though the above intermolecular channel was fast delivered to the metal center of the cocatalyst, working out the most critical challenge in the oriented electron delivery from CPs to cocatalyst and achieves the efficient π-conjugation system for $CO_2$ photoreduction (Supplementary Fig. 39)[12].

## Discussion

To summarize, we found that an intermolecular cascaded channel for the electron delivery can be established through the modification of intramolecular and intermolecular π–π conjugation. Encouragingly, with the strategy of utilizing the Suzuki-Miyaura coupling instead of Sonogashira-Hagihara coupling to modify both intramolecular conjugation and intermolecular interaction of CPs, the photoelectrons generated from the pyrene part were localized on the benzene part of CPs, which in turn deliver the photoelectrons faster to Co (II) bipyridine complexes through the intermolecular cascaded channel. The current work reported directly connected net-like CPs systems exhibits highest CO evolution activity of 2247 $\mu$mol g$^{-1}$ h$^{-1}$ with an apparent quantum efficiency exceeds 3.39% among all reported CPs and most considerable enhancement of 138-times compared to unmodified CPs (N-CP-A) in $CO_2$ photoreduction with adding Co (II) bipyridine complexes as cocatalyst. From the theoretical calculations and transient spectroscopy techniques, this high efficiency could be attribute to the intermolecular cascaded channels built by modification of π-conjugation in CPs. In addition, this strategy constructs a reliable system of $CO_2$ photoreduction over CPs via smart engineering in molecular level, which figures out the most critical challenge in the oriented electron delivery as well as providing a viable route for designing high-efficiency polymers-based systems of $CO_2$ photoreduction.

## Methods

**Synthesis of CP-A series (L-CP-A and N-CP-A).** All reagents were purchased from Sigma-Aldrich or Tokyo Chemical Industry without further purification. The CP-A series were synthesized according to Sonogashira-Hagihara cross-coupling polycondensation. In details, a dry 250 mL round-bottom flask was charged with two monomer reactants, $Pd(PPh_3)Cl_2$ and CuI, and mixed solvent of dimethyl formamide/triethylamine (DMF/TEA). The mixture was degassed by bubbling with Ar for 30 min, and then the resulting mixture was stirred at 80 °C for 24 h under Ar condition. After that, the precipitate was collected by filtration, and the solid was washed with methanol and $CH_2Cl_2$ in the Soxhlet for 48 h. The final product was dried at 60 °C overnight. For synthesis of L-CP-A, 1,4-diethynylbenzene (189 mg, 1.5 mmol), 1,6-dibromopyrene (540 mg, 1.5 mmol) $Pd(PPh_3)Cl_2$ (27 mg), CuI (5 mg), DMF (60 mL), and TEA (60 mL) were used. For N-CP-A, 1,4-diethynylbenzene (189 mg, 1.5 mmol), 1,3,6,8-Tetrabromopyrene (388 mg, 0.75 mmol), $Pd(PPh_3)Cl_2$ (27 mg), CuI (5 mg), DMF (60 mL), and TEA (60 mL) were used.

**Synthesis of CP-D series (L-CP-D and N-CP-D).** All reagents were purchased from Sigma-Aldrich or Tokyo Chemical Industry without further purification. The CP-D series were synthesized according to Suzuki-Miyaura cross-coupling polycondensation. In details, a dry 250 mL round-bottom flask was charged with two monomer reactants, $Pd(PPh_3)_4$, $K_2CO_3$, and mixed solvent of dimethyl formamide/water (DMF/$H_2O$). The mixture was degassed by bubbling with Ar for 30 min and then the resulting mixture was stirred at 150 °C for 24 h under Ar condition. After that, the precipitate was collected by filtration, and the solid was washed with methanol and $CH_2Cl_2$ in the Soxhlet for 48 h. The final product was dried at 60 °C overnight. For synthesis of L-CP-D, 1,4-phenylenediboronic acid (248 mg, 1.5 mmol), 1,6-dibromopyrene (540 mg, 1.5 mmol), $Pd(PPh_3)_4$ (10 mg), $K_2CO_3$ (2.0 g) DMF (60 mL), and $H_2O$ (8 mL) were used. For N-CP-D, 1,4-phenylenediboronic acid (248 mg, 1.5 mmol), 1,3,6,8-Tetrabromopyrene (388 mg, 0.75 mmol), Pd$(PPh_3)_4$ (10 mg), $K_2CO_3$ (2.0 g) DMF (60 mL), and $H_2O$ (8 mL) were used.

**Characterization.** Transmission electron microscopy (TEM, Tecnai G2 F20, FEI, Holland) and scanning electron microscope (SEM, SU8010, Hitachi, Japan) were used to analyze the morphologies of polymers. The diffractometer (D8 advance, Bruker, Germany) with Cu Kα radiation was used to record the Powder X-ray diffraction (PXRD). The solid-state $^{13}C$ cross polarization magic angle spinning ($^{13}C$-CP/MAS) NMR spectra (Avance III HD 400 MHz spectrometer, Bruker, Germany) were measured to analyze the structure of polymers. The infrared spectra were recorded using a Fourier transform-infrared (FT-IR) spectrometer (Nicolet 6700, Thermo Scientific, USA). Raman spectra were collected using a Raman spectrometer (DXR, Thermo Scientific, USA). The low-pressure gas adsorption measurements were investigated with a gas adsorption analyzer (ASAP 2040, Micrometrics, USA) using nitrogen/carbon dioxide as the adsorbate at 77 K. UV–Vis diffused reflectance spectra (DRS) were measured on a spectrometer (UV-3100, Shimadzu, Japan). The thermal stability was evaluated by thermal gravity analysis (TGA, TG 209 F3 Tarsus, Netzsch, Germany) with the temperature increased to 10 °C min$^{-1}$ under air atmosphere. Metal content was determined by inductively coupled plasma mass spectrometry (ICP-MS, 7800, Agilent Technologies, USA), where the sample was first digested by $H_2SO_4$/$HNO_3$ (0.8 mL/0.2 mL) solvent at 60 °C. The surface electronic states of polymers were analyzed via X-ray photoelectron spectroscopy (XPS, ESCALAB 250Xi, Thermo Scientific, USA). Cyclic voltammetry (CV) measurements, electrochemistry impedance spectroscopy (EIS) and photocurrent intensity response measurements were performed in a typical three-electrode cell system. Electrochemical measurements were carried out in a conventional three-electrode system using an Ag/Ag$^+$ as reference electrode, platinum plate as the counter electrode, and the sample onto glassy carbon as the working electrode. The electrochemical workstation was a CHI 660E potentiostat (Shanghai Chenhua Co.) and CVs were collected at a scan rate of 50 mV s$^{-1}$ with the protection of nitrogen. A solution of 0.1 M TBAPF$_6$ in $CH_3CN$ was used as the electrolyte.

**FP-TRMC measurement.** FP-TRMC measurements were carried out at room temperature under a $N_2$ atmosphere, using different CPs powder on poly (methylmethacrylate) (PMMA) films. The films were cast onto quartz substrates. The microwave power and frequency were set at 3 mW and ~9.1 GHz, respectively. Charge carriers were generated in the films by direct excitation of CPs using third-harmonic generation ($\lambda = 355$ nm) light pulses from a Nd: YAG laser (Spectra Physics, INDI-HG). The excitation density was tuned at $1.8 \times 10^{16}$ photons cm$^{-2}$. The TRMC signal from a diode was recorded on a digital oscilloscope (Tektronix, TDS 3032B). Comparison of the integrated photocurrents with the polymer standard (poly-9,9′-dioctylfluorene, $\phi \sim 2.3 \times 10^{-4}$) allowed determination of the quantum efficiency of charge carrier generation for the CPs samples. The local-scale charge carrier mobility Σμ was estimated by the quotient of ($\phi\Sigma\mu$) $_{max}$ by $\phi$.

**Photocatalytic activities measurement.** The photoreduction $CO_2$ activities of all CPs were carried out in gas-closed system with a gas-circulated pump. The setup of the photocatalytic system is illustrated in Fig. S17. In detail, the 5 mg catalyst, 50 mL of solution (acetonitrile/water = 7:3), 5 mL of TEOA and the synthesized Co (II) bipyridine complexes cocatalyst[43] were added in a Pyrex glass reaction cell which was connected to the $CO_2$ reduction system. After complete evacuation of the reaction system (no $O_2$ or $N_2$ could be detected by gas chromatography), ~80 kPa of pure $CO_2$ gas was injected into the airtight system. After adsorption equilibrium, a 300 W xenon lamp (~100 mW/cm$^2$) with a UV-cut filter (L42), to

remove light with wavelengths lower than 420 nm ($\lambda > 420$ nm) was used as the light source. The produced $H_2$ and CO was analyzed by two gas chromatographs (GC-8A and GC-2014, Shimadzu Corp., Japan) equipped with different chromatographic column[12,22].

**Isotope labeling measurement**. The isotope labeling measurement was carried out by using $^{13}CO_2$ gas (Isotope purity, 99% and chemical purity, 99.9%, Tokyo Gas Chemicals Co., Ltd.) instead of pure $^{12}CO_2$ gas (Chemical purity, 99.999%, Showa Denko Gas Products Co., Ltd.) as the carbon source with the same reaction set as mentioned above and the gas products were analyzed by gas chromatography-mass spectrometry (JMS-K9, JEOL-GCQMS, Japan and 6890N Network GC system, Agilent Technologies, USA) equipped with two different kinds of column for detecting the products of $^{13}CO$ (HP-MOLESIEVE, 30 m × 0.32 mm × 25 μm, Agilent Technologies, USA) and source of $^{13}CO_2$ (HP-PLOT/Q, 30 m × 0.32 mm × 20 μm, Agilent Technologies, USA), respectively[12,22].

**In-situ DRIFTS measurement**. In-situ DRIFTS measurement were carried out by FT-IR spectrometer (Nicolet iS50 Thermo Scientific, USA) with a designed reaction cell simulated in Fig. S23a. The substrate lying in the center of the designed reaction cell and a thin layer of N-CP-D mixture with or without cocatalyst as the model sample was placed uniformly on the substrate. An ultra-high vacuum pump was used to pump out all the gases in the reaction cell and adsorbed on the photocatalyst surface. Then the large amount of carbon dioxide or carbon monoxide was pumped in to construct a $CO_2$ or CO atmosphere for $CO_2$ photoreduction or CO adsorption, respectively. At last, visible light was turned on and the IR signal was in-situ collected through MCT detector along with the reaction.

**TA measurement**. The output pulses of a 1 kHz Ti: sapphire regenerative amplifier (Solstice, Spectra-Physics) were split into two beams. One of the beams was used as an excitation light source of an optical parametric amplifier (OPA) (TOPAS prime, NIR-UV-Vis. LIGHT CONVERSION Inc.). The 420 nm output of the OPA was as the excitation light source. The excitation light was chopped at 500 Hz by an optical chopper. The other beam was used as an excitation light source of another OPA. The output of the OPA was used as the wavelength-tunable probe and reference light source (500–2600 nm). Si photodetectors and InGaAs photodetectors were used to detect the probe and reference light depending on probe light wavelength (500–1000 nm and 1000–2600 nm respectively). (Supplementary Fig. 40).

**TR-PL measurement**. Photoluminescence spectra decay curves were obtained by using a Hamamatsu instrument (Hamamatsu C5680, Japan) with a 1 kHz Ti: sapphire regenerative amplifier (Solstice, Spectra-Physics) was used as an excitation light source of an OPA (TOPAS prime, NIR-UV-Vis. LIGHT CONVERSION Inc.). The CPs were dispersed in the solvent of acetonitrile and water solution with scavenger of TEOA. The argon was firstly pumped into a sealed cell for 40 min by using a pipe inlet to exclude the dissolved oxygen and other atmosphere. After the above pre-treatment process, the TR-PL signals were recorded and these samples are named L-CP-A in Ar, L-CP-D in Ar, N-CP-A in Ar and N-CP-D in Ar, respectively. As a comparison, the $CO_2$ was subsequently pumped into this sealed cell for 40 min by using a pipe inlet. The TR-PL signals were also recorded, and these samples are named L-CP-A in $CO_2$, L-CP-D in $CO_2$, N-CP-A in $CO_2$ and N-CP-D in $CO_2$, respectively.

## Data availability

The data that support the plots within this paper and other findings of this study are available from the corresponding author upon reasonable request.

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

## Acknowledgements

This work received financial support from the National Special Key Project for Transgenic Breeding (No. 2016ZX08001001), the National Science Foundation of China (Nos. 21633004, 51872107, 51902121, 51572101, 21607047, and 21502059), the World Premier International Research Center Initiative (WPI Initiative) on Materials Nanoarchitec-
tonics (MANA), MEXT (Japan), KAKENHI (18H02065), MEXT, and Photoexcitonix Project in Hokkaido University, Japan, Fundamental Research Funds for the Central Universities of China (Nos. 2015PY120, 2015PY047, 2016PY088, and 2018QD011), and the Natural Science Foundation of Hubei Province (Nos. 2019CFB322, and 2016CFB193).

## Author contributions

S.W., X.H. X.D., and S.J. contributed equally to this work. J.Y. and S.W. conceived the idea, designed the experiments. S.W. fabricated the materials. S.W. and X.H. performed the photocatalytic experiment. F.I. and Y.W. conducted the TA and TR-PL measurement, P.W. and M.O. completed the theoretical computations, S.S., W.M., and J.M. conducted the FP-TRMC measurements, X.D., B.J., and S.J. performed the in-situ DRIFTS, XRD, and DRS measurements, X.W. and Y.X. carried out the NMR, ICP-MS and XPS tests, Y.L. and H.H. conducted the $CO_2/N_2$ adsorption, TG, TEM, and SEM measurements, X.M. performed the isotopic experiments. S.W., X.H., X.D., and S.J. wrote the manuscript. J.Y. and H.C. supervised the work and revised the manuscript. All authors contributed to discussion of the results and the manuscript preparation.

## Competing interests

The authors declare no competing interests.
