## [Peer Review File · Nature Communications]

Reviewers' comments:

Reviewer #1 (Remarks to the Author):

This manuscript developed the construction of intermolecular cascaded π -conjugation channels by modifying both intramolecular and intermolecular conjugation to optimize the photogenerated electron delivery and improve the performance of CO₂ photoreduction. The highest CO evolution activity of 2247 $\mu\text{mol g}^{-1} \text{h}^{-1}$ among all reported conjugated polymers has been achieved. These results may be interesting and inspire scientists in the field to further improve the performance of CO₂ photoconversion. However, there are some issues on data analyses about structure and properties. Therefore, revision on this draft is necessary before acceptance for publication.

1. The discussion of the band structure is too superficial. Please added the detailed investigation in the manuscript, such as Mott-Schottky test and UPS spectra.
2. Page 8, the cyclic-voltammetry method and EPR measurement cannot demonstrated that the electrons on LUMO of CPs can have the ability to produce Co (I) from Co (II), because of solvent effect. The test conditions for EPR measurement should be provided. More convincing evidence for the existence of Co (I) should be provided, e.g. in-situ XPS.
3. According to supplementary Table S4, residual palladium (0.661 wt.%) retains in CPs matrix. Whether the residual Pd affect the photocatalytic CO₂ reduction activity? There is no explanation about it in the manuscript.
4. In Fig. 4a, CO₂ absorption-desorption isotherms curves were employed to demonstrate that the capacity for CO₂ adsorption was not the determining factor for the CO₂ photoreduction. However, the chemisorption of CO₂ is a more important factor in CO₂ photoreduction. The chemisorption of CO₂, e.g. CO₂-FTIR, should be added.
5. In Fig. 4c and supplementary Table S6, the author tried to highlight the fact that the photoexcited electrons can be fast delivered to CO₂ from catalyst. Nevertheless, the short lifetime (τ_1) was also shortened in CO₂ atmosphere. The average lifetime and experimental details of TRPL should be provided.
6. The decomposition of the CP should be considered on the conversion of CO₂ to CO.
7. The photocatalytic stability should be carried out and the mechanism of the apparent degradation of the photocatalytic activity should be discussed.

Reviewer #2 (Remarks to the Author):

I am supportive of this work as this is very topical area of research and a range of interesting measurements (such as TRMC and TAS) are used to gain understanding and the observed AQYs are high, which is also important in this field.

Having said this there are a couple of flaws which I believe should be addressed before this can be accepted:

- I strongly disagree with the premise of 'molecular architecture' or design. I actually think it makes the manuscript weaker as there is a risk that other important factors have not been considered, the data set is simply too small to draw strong conclusions, in particular in photocatalysis where many different processes are taking place. I am also a little unsure about the notion of improving alkynyl materials: As far as I can tell these have not been previously reported and based on what has been reported for hydrogen production from water, I would not have expected them to act as particularly

good photocatalysts for CO₂ reduction. I would recommend focusing on reporting these materials and dropping the design aspect (including the title), as it is questionable in this field
- Transient absorption spectroscopy can be very insightful to understand the dynamics in these materials. However, it would be much more useful to perform the measurement in the presence of scavenger and/or CO₂ to understand what is happening under photocatalysis conditions.

A few other more minor comments and questions:

- 1) At first I was very excited about these results only to see later that TEOA was used. This is ok at this stage, but the manuscript needs to state clearer in the main-text that a scavenger was used.
- 2) Net-like materials are usually referred to as conjugated microporous polymers in the materials community.
- 3) Comparing rates to other reports makes little sense as this ignores the fact that set-ups are very different and light sources vary (statement: 'highest CO evolution efficiency' and in a few other places). It would be much better to compare AQYs, which are already in the manuscript, to other reports.
- 4) Particle size and dispersion in the photocatalysis mixture are currently not considered. Some reports indicate that this might be important and should be measured and added.
- 5) Important experimental details are missing: What was the pressure of the photolysis experiment? This needs to be added to figure captions and also mentioned in the main-text. How much of the Co complex was used? I was a little unsure at times and it needs to be stated clearer.
- 6) CO₂ absorption of a dry powder can be very different compared to a measurement under wet conditions. This has been reported for conjugated microporous polymers and should be measured if it is considered to be important.
- 7) Selectivity is only briefly discussed in the main-text but omitted from Fig.3. I am a little unsure how important it is right now as the area is in its infancy, but it seems to be standard practise in the field and should be added.
- 8) Were the TR-PL measurements in Fig.4 performed in the presence of a scavenger?

Reviewer #3 (Remarks to the Author):

The manuscript of J.H. Ye et al. reports on the original construction of intermolecular cascaded n -conjugation channels for powering CO₂ photoreduction by modifying both intramolecular and intermolecular conjugation of conjugated polymers. This study is of great interest and reports a rather novel approach. Detailed experimental analysis and theoretical calculations were conducted to verify the proposed approach and conclusion. This work might interest those in pursuit of better photogenerated electron delivery from the viewpoint of the molecular architecture. Though I inclined to recommend the publication with Nat. Commun., there are also some important revisions need to be made:

1. Could you please give the loading content of the Co (II) bipyridine complexes cocatalyst on the CPs as this may be critical for the photocatalytic efficiency?
2. As the author showed in Table S4, there are some residual palladium and copper metals detected in the CPs. Please explain if these metals, especially the Pd, will have any activity in the photocatalytic process.
3. In this study, the LUMO and HOMO were obtained by hybrid density-functional-theory-based first-principles calculations. The experimental verifications for these values by UPS or Mott-Schottky plot should be conducted.
4. Authors mentioned that "The conductivity transients and calculated charge mobilities for CPs are displayed in Fig. 2a, in which the L-CP-A, with a linear structure and alkynyl group, exhibits charge mobility (μ_{tot}) of 0.32 cm² V⁻¹ s⁻¹ ($\varphi \Sigma\mu = 7.4 \times 10^{-5}$ V⁻¹ s⁻¹). As expected, the charge mobility of L-CP-D in the absence of alkynyl decreased to a much lower value of 0.15 cm² V⁻¹ s⁻¹ ($\varphi \Sigma\mu = 3.4 \times 10^{-5}$ V⁻¹ s⁻¹)." (Page 6). However, there is no corresponding results shown in Fig.2a. Please check this carefully.
5. Authors claimed that "Besides, the net-like CPs (N-CP-A and N-CP-D) possess more cocatalyst

absorption sites of phenyl in the units, thus resulting in enhanced performance of net-like CPs (N-CP-A and N-CP-D) than those of linear CPs (L-CP-A and L-CP-D)" (Page 9). As Fig.4a shows, the N-CPs exhibited greater CO₂ adsorption than the L-CPs. Does the CO₂ adsorption have any effects on the CO₂ photoreduction efficiency except for the cocatalyst absorption sites number? This may be a result of multi-factors.

6. There is a clerical error in SI (page 46). "As shown in Figure S4" should be "As shown in Figure S34".

Then, I would highly suggest these minor revisions to be thoroughly addressed before re-submission.

Reviewers' comments:

Reviewer #1 (Remarks to the Author):

This manuscript developed the construction of intermolecular cascaded π -conjugation channels by modifying both intramolecular and intermolecular conjugation to optimize the photogenerated electron delivery and improve the performance of CO₂ photoreduction. The highest CO evolution activity of 2247 $\mu\text{mol g}^{-1} \text{h}^{-1}$ among all reported conjugated polymers has been achieved. These results may be interesting and inspire scientists in the field to further improve the performance of CO₂ photoconversion. However, there are some issues on data analyses about structure and properties. Therefore, revision on this draft is necessary before acceptance for publication.

Response: The authors thank the reviewer for the valuable comments. These comments are very helpful for improving the quality and value of this article.

1. The discussion of the band structure is too superficial. Please added the detailed investigation in the manuscript, such as Mott-Schottky test and UPS spectra.

Response: Based on previous reports, the energy band structures of these CPs were investigated via cyclic voltammetry measurement, which was regarded as the one of most suitable way to determine the LUMO and HOMO levels of these conjugated polymers (Adv. Mater. 2015, 27, 6265; Angew. Chem. Int. Ed. 2015, 54, 13594; Angew. Chem. Int. Ed. 2016, 55, 9783; Angew. Chem. Int. Ed. 2016, 55, 9202;). Combining with the bandgap energy from diffuse reflectance spectrum (DRS), the energy levels of these CPs were precisely positioned and showed in the Supplementary Table S1. According to the reviewer's suggestion, we also conducted the Mott-Schottky test (Figure R1) and UPS spectra (Figure R2). to further explore the band structures of these CPs that are coincidence well with our cyclic voltammetry (CV) results..

Figure R1. Mott-Schottky plots for CPs in 0.1 M Na₂SO₄ at pH=7 by using Ag/AgCl with saturated KCl as reference electrode.

As shown in Figure R1. The flat-bands of L-CP-D, L-CP-A, N-CP-D and N-CP-A were estimated to be -1.35, -0.76, -1.22, -0.65 V versus the Ag/AgCl (-1.15, -0.56, -1.02, -0.45 V versus normal hydrogen electrode), respectively. The flat-band potential usually exhibits a potential difference of 0-0.1 V to the LUMO levels (Nat. Mater. 2019, 18, 827), the LUMO of L-CP-D, L-CP-A, N-CP-D and N-CP-A were determined to be -1.25, -0.66, -1.12, -0.55 V versus normal hydrogen electrode, respectively. With the assistant of band gap from DRS measurement, the energy levels of these CPs were showed in Table R1, which showed high accordance with the energy levels determined by cyclic voltammetry measurement.

Table R1. The energy levels within the L-CP-D, L-CP-A, N-CP-D and N-CP-A obtained from the Mott-Schottky test.

	L-CP-D	N-CP-D	L-CP-A	N-CP-A
Band gap	2.62 eV	2.47 eV	2.10 eV	1.98 eV
HOMO	1.37 V	1.35 V	1.44 V	1.43 V
LUMO	-1.25 V	-1.12 V	-0.66 V	-0.55 V

Figure R2. UPS spectra in the cutoff and the onset energy regions of L-CP-D, L-CP-A, N-CP-D and N-CP-A.

As shown in Figure R2, ultraviolet photoelectron spectroscopy (UPS) was employed to evaluate the energy levels of CPs. The cutoff (E_{cutoff}) and onset (E_i) energy regions are shown in the UPS spectra, respectively. According to the equation $\phi = 21.2 - (E_{\text{cutoff}} - E_i)$, the HOMO levels of the L-CP-D, L-CP-A, N-CP-D and N-CP-A were calculated to be 1.34, 1.23, 1.46, 1.45 V versus normal hydrogen electrode, respectively. With the assistant of band gap analysis from DRS measurement, the energy levels of CPs were showed in Table R2, which showed high accordance with the energy levels determined by cyclic voltammetry measurement and Mott-Schottky test.

Table R2. The energy levels within the L-CP-D, L-CP-A, N-CP-D and N-CP-A obtained from the UPS test.

	L-CP-D	N-CP-D	L-CP-A	N-CP-A
Band gap	2.62 eV	2.47 eV	2.10 eV	1.98 eV
HOMO	1.34 V	1.23 V	1.46 V	1.45 V
LUMO	-1.28 V	-1.24 V	-0.64 V	-0.53 V

In summary, we have revised the discussion of the band structure in the Manuscript as follow:

“Cyclic voltammetry (CV) measurements were also conducted, the HOMO position can be determined by the irreversibility of the oxidation peaks due to the irreversible oxidation process of

the CPs at the impressed voltage (Supplementary Fig. 10) revealed different energy levels within the CPs (Supplementary Table 1)³⁹. In addition, their energy levels were further investigated by the UPS (Ultraviolet Photoelectron Spectroscopy) (Supplementary Fig. 11) and Mott-Schottky test (Supplementary Fig. 12), which showed a high accordance with the energy levels determined by CV measurement (Supplementary Table 2).”

2. Page 8, the cyclic-voltammetry method and EPR measurement cannot demonstrate that the electrons on LUMO of CPs can have the ability to produce Co (I) from Co (II), because of solvent effect. The test conditions for EPR measurement should be provided. More convincing evidence for the existence of Co (I) should be provided, e.g. in-situ XPS.

Response: We thank the reviewer for this comment. We obtained the N-CP-D powder with Co complexes by a natural adsorption process and then place it into the EPR sample tube. We used a degassing device to make sure that tube is oxygen-free and seal the sample tube by sintering the nozzle. After these careful pre-treatments, we tested the EPR spectra of Co before and after light illumination. Through the above mentioned method, we cannot find a peak that can be assigned to the Co (I) because of its low-spin state of the Co (I). Only a weak peak assigned to Co (II) were observed in the EPR measurement. However, based on the previous reports (J. Mol. Catal. A-Chem. 2013, 193, 27; Angew. Chem. Int. Ed. 2016, 55, 14310; ChemSusChem 2019, 12, 4493), the in-situ generated Co (I) are widely recognized as the major active species for CO₂ reduction. According to reviewer’s suggestion, we also conducted the in-situ XPS measurement to explore the existence of Co (I) (Figure R3).

Figure R3. In-situ XPS spectra of N-CP-D with Co complexes (a) and the corresponding multi peaks separation spectra (b).

As shown in Figure R3, compared to the N-CP-D without irradiation, the binding energy of Co 2p in N-CP-D with irradiation shifted to a lower binding energy. From the further multi peaks separation process, two new peaks appear Co 2p_{3/2} and Co 2p_{1/2} that show lower binding energy than that of Co²⁺, clearly revealing the photo-induced generation of Co⁺ in N-CP-D.

We have added the following sentences for the EPR measurement and the description of the photo-induced generation of Co⁺ both in Supplementary information and Manuscript.

In Manuscript:

“We also demonstrated that photo-excited electrons on the LUMO of CPs in the present work do have the ability to reduce Co (II) bipyridine complexes to Co (I) bipyridine complexes via the cyclic-voltammetry spectrum (Supplementary Fig. 19). In addition, the weakened EPR signal (Supplementary Fig. 20, see Methods for experimental details)²² indicates the weakening of high-spin-state Co (II) upon visible-light irradiation that maybe reduced to low-spin-state Co (I). In order to get more convincing evidence for the existence of the low-spin-state of Co (I), the in-situ XPS (Supplementary Fig. 21) were performed over N-CP-D. It showed a lower binding energy than that of Co²⁺, which clearly revealed the existence of photo-induced low-spin-state Co⁺ in this system.”

In Supplementary information:

“**EPR measurement.** N-CP-D powder with Co complexes was obtained by an adsorption process and then place it into the EPR sample tube. A degassing device was used to make the tube oxygen-free and seal the sample tube by sintering the nozzle. After these pre-treatments, the sample was tested before and after irradiation with an electron spin resonance (JES-FA200, JEOL, Japan) spectrometer.”

3. According to supplementary Table S4, residual palladium (0.661 wt.%) retains in CPs matrix. Whether the residual Pd affect the photocatalytic CO₂ reduction activity? There is no explanation about it in the manuscript.

Response: Indeed, there are trace amounts of residual palladium retains in all CPs. Despite it is inevitable that trace amounts of palladium remained in the CPs, all CPs without adding Co

complexes as cocatalyst showed negligible photocatalytic CO₂ reduction activity. After adding cocatalyst, the CP-A series still exhibited low activities in CO₂ reduction, while the CP-D series showed a great enhancement, revealing the enhanced activities are attributed to the intermolecular electron transfer between CP-D series and Co complexes.

We have added the following sentences for the explanation that the residual Pd has negligible effect on the photocatalytic CO₂ reduction activity in Manuscript.

“All CPs showed quite low activity in the absence of cocatalyst, as well as the CP-D series in the presence of isolated cobalt chloride or dipyriddy (Supplementary Fig. 22), implying that the residual Pd has negligible effect on the photocatalytic CO₂ reduction activity (Supplementary Table 5).”

4. In Fig. 4a, CO₂ adsorption-desorption isotherms curves were employed to demonstrate that the capacity for CO₂ adsorption was not the determining factor for the CO₂ photoreduction. However, the chemisorption of CO₂ is a more important factor in CO₂ photoreduction. The chemisorption of CO₂, e.g. CO₂-FTIR, should be added.

Response: We thank the reviewer for this comment. The CO₂ absorption of a dry powder is very different compared to measurement under wet conditions based on the previous report (J. Am. Chem. Soc. 2012, 134, 10741). The chemisorption of CO₂ is a more important factor than CO₂ adsorption in CO₂ photoreduction. We therefore employed the in-situ infrared technology to investigate the adsorption and chemisorption of CO₂ over CPs in a solvent-containing environment (Figure R4).

Figure R4. In-situ FT-IR spectra of N-CP-D with Co complexes for the adsorption of CO₂ (a) and the chemisorption of CO₂ (b) in a solvent-containing environment.

As shown in Figure R4, although the CP-A series exhibited greater CO₂ adsorption than the CPs-D series, the L-CP-D and N-CP-D showed an enhanced intensity both in the region of CO₂ adsorption (a) and the region of CO₂ chemisorption (b) than L-CP-A and N-CP-A. It suggests that the CO₂ adsorption capacity of CPs-D series is stronger than that of CPs-A series under the solvent-containing environment, which provides favorable conditions for the subsequent CO₂ reduction reaction.

We have revised the following sentence in Manuscript for the explanation of the CO₂ adsorption.

“Nevertheless, the CO₂ absorption of a dry powder is very different from the wet conditions⁵⁰, the in-situ FT-IR indicates L-CP-D and N-CP-D exhibit enhancement both in CO₂ adsorption and CO₂ chemisorption than L-CP-A and N-CP-A (Supplementary Fig. 30). It means that the CO₂ adsorption capacity of CPs-D series is stronger than that of CPs-A series under the solvent-containing environment, which provides favorable conditions for the subsequent CO₂ reduction reaction.”

50 Dawson, R. *et al.* Impact of water coadsorption for carbon dioxide capture in microporous polymer sorbents. *J. Am. Chem. Soc.* **134**, 10741-10744 (2019).

5. In Fig. 4c and supplementary Table S6, the author tried to highlight the fact that the photoexcited electrons can be fast delivered to CO₂ from catalyst. Nevertheless, the short lifetime (τ_1) was also shortened in CO₂ atmosphere. The average lifetime and experimental details of TRPL should be provided.

Response: We thank the reviewer for this comment. We have added the average lifetime in Table R3 and the experimental details of TR-PL in the Supplementary Information.

“**TR-PL measurement.** Photoluminescence spectra decay curves were obtained by using a Hamamatsu instrument (Hamamatsu, Japan) with a 1kHz Ti: sapphire regenerative amplifier (Solstice, Spectra-Physics) was used as an excitation light source of an optical parametric amplifier (OPA) (TOPAS prime, NIR-UV-Vis. LIGHT CONVERSION Inc.). The CPs were dispersed in the solvent of acetonitrile and water solution with scavenger of triethanolamine. The

argon was firstly pumped into a sealed cell for 40 min by using a pipe inlet to exclude the dissolved oxygen and other atmosphere. After the above pre-treatment process, the TR-PL signals were recorded and these samples are named L-CP-A in Ar, L-CP-D in Ar, N-CP-A in Ar and N-CP-D in Ar, respectively. As a comparison, the CO₂ was subsequently pumped into this sealed cell for 40 min by using a pipe inlet. The TR-PL signals were also recorded and these samples are named L-CP-A in CO₂, L-CP-D in CO₂, N-CP-A in CO₂ and N-CP-D in CO₂, respectively.”

As shown in Table R3, it was indeed mentioned by the reviewer, the short lifetime (τ_1) was also shortened in a CO₂ atmosphere. It can be mainly attributed to a large number of photogenerated electrons which can be rapidly delivered to CO₂ via the cocatalyst and only a small number of electrons with a short lifetime involved in the detectable recombination process of TR-PL under CO₂ atmosphere.

We can find that the average lifetime (τ_{av}) of L-CP-D and N-CP-D also showed an obvious decrease in CO₂ atmosphere than that in an argon atmosphere, while the average lifetimes of L-CP-A and N-CP-A exhibited no obvious change either in CO₂ or in argon. We thus attribute these differences to the fact that electrons can be fast delivered to CO₂ via the cocatalyst in L-CP-D and N-CP-D.

Table R3. Parameters of the time-resolved photoluminescence decay curves according to a biexponential decay in different atmosphere.

	In argon atmosphere					In CO ₂ atmosphere				
	τ_1 (ns)	A ₁ (%)	τ_2 (ns)	A ₂ (%)	τ_{av} (ns)	τ_1 (ns)	A ₁ (%)	τ_2 (ns)	A ₂ (%)	τ_{av} (ns)
L-CP-D	0.242	45.8	2.606	54.2	1.523	0.126	78.6	0.840	21.4	0.279
N-CP-D	0.208	35.3	3.462	64.7	2.313	0.168	82.2	1.316	17.8	0.372
L-CP-A	0.105	83.8	0.792	16.2	0.216	0.123	85.2	0.941	14.8	0.244
N-CP-A	0.107	84.3	0.881	15.7	0.229	0.124	84.9	1.023	15.1	0.259

We also revised the description of TR-PL in the Manuscript as follows.

“For both N-CP-D (Fig. 4c) and L-CP-D (inset of Fig. 4c), the average lifetime of electrons (Supplementary Table 8) in a CO₂ atmosphere was significantly shortened compared to that in an

argon atmosphere, which can be mainly attributed to the large amount of photogenerated electrons can be fast delivered to CO₂ via the cocatalyst and only a small number of electrons with short lifetime involved into the detectable recombination process under the CO₂ atmosphere (Supplementary Fig. 34), which can be further confirmed by the TA spectra in different atmosphere (Supplementary Fig. 35).”

6. The decomposition of the CP should be considered on the conversion of CO₂ to CO.

Response: We thank the reviewer for this comment. To avoid possible decomposition of CPs by the photo-induced thermal effects to CO, we kept the system temperature during the reaction from being too high through a circulating cooling system. By using this condition, the CPs form decomposition by the photo-induced thermal effects can be effectively avoided as indicated by the isotopic experiment.

From the isotopic experiment, the ratio of generated ¹³CO to ¹²CO was found to be also the same as the ratio of ¹³CO₂ to ¹²CO₂ in the source gas, indicating that almost all produced ¹³CO indeed originated from ¹³CO₂ in the CO₂ photoreduction over the CPs while not the decomposition product. In addition, the CO₂ reduction experiment of CPs without adding Co complexes shows negligible activity in CO generation, which indicates the decomposition over CPs during the CO₂ reduction is negligible.

7. The photocatalytic stability should be carried out and the mechanism of the apparent degradation of the photocatalytic activity should be discussed.

Response: We have carried out the photocatalytic stability showed in Figure 3c, from which there is only a slight decrease in the fifth cycle and the photocatalytic activity remains to be 91% of the original cycle. We attribute this slight decrease to the fresh Co complexes which are not added to the system in the photocatalytic stability measurement.

To better clarify this point, we also added the discussion about the slight decrease of the photocatalytic activity in the Manuscript as follows.

“Moreover, the stability test over L-CP-D and N-CP-D indicates that after 5 cycles, the CO evolution was still high (maintaining 91% to the original cycle without adding fresh Co complexes

to the system) as compared to the initial values (Fig. 3c), suggesting the adequate stability of these CPs (Supplementary Fig. 25).”

Reviewer #2 (Remarks to the Author):

I am supportive of this work as this is very topical area of research and a range of interesting measurements (such as TRMC and TAS) are used to gain understanding and the observed AQYs are high, which is also important in this field.

Response: The authors thank the reviewer for the valuable comments. These comments are very helpful for improving the quality and value of this article.

Having said this there are a couple of flaws which I believe should be addressed before this can be accepted:

- I strongly disagree with the premise of ‘molecular architecture’ or design. I actually think it makes the manuscript weaker as there is a risk that other important factors have not been considered, the data set is simply too small to draw strong conclusions, in particular in photocatalysis where many different processes are taking place. I am also a little unsure about the notion of improving alkynyl materials: As far as I can tell these have not been previously reported and based on what has been reported for hydrogen production from water, I would not have expected them to act as particularly good photocatalysts for CO₂ reduction. I would recommend focusing on reporting these materials and dropping the design aspect (including the title), as it is questionable in this field.

Response: Thank you very much for your helpful suggestion. According to your insightful suggestion, we have revised the title into “Intermolecular Cascaded π -Conjugation Channels for Electron Delivery Powering CO₂ Photoreduction”, which weakened the design aspect in the title. Besides, we agree that photocatalysis is a very complicated reaction where many different processes are taking place. So, we do our best to weaken the concept of molecular architecture in the Manuscript and highlight that intermolecular electron delivery is one of the critical factors for CO₂ reduction, which has never been reported in previous research.

About the notion of improving alkynyl materials, the alkynyl materials are reported as the candidates for hydrogen production from water better than materials without alkynyl (J. Catal.

2017, 350, 64). But, the materials without alkynyl showed significant enhancement in CO₂ reduction, which acts as good photocatalysts for CO₂ reduction. We attribute this difference in two similar photocatalytic reactions to the different kinds of cocatalyst. In hydrogen production reaction, the Pt loaded on the surface of materials as the cocatalyst and the photogenerated electrons need to be transferred to cocatalyst across the materials. However, the Co complexes act as the cocatalyst in the CO₂ reduction reaction, which is the independent molecule to the materials. The absence of alkynyl in the material concentrates the photogenerated electron to promote the intermolecular electron delivery.

- Transient absorption spectroscopy can be very insightful to understand the dynamics in these materials. However, it would be much more useful to perform the measurement in the presence of scavenger and/or CO₂ to understand what is happening under photocatalysis conditions.

Response: We thank the reviewer for this comment. We have conducted the transient absorption spectroscopy over N-CP-D in a solvent-containing environment in the presence of scavenger. We have compared the different dynamics between the atmosphere of CO₂ and argon in Figure R5.

Figure R5. Kinetics of electrons in transient absorption over N-CP-D in the presence of scavenger under the atmosphere of CO₂ or argon with a probe wavelength of 2200 nm. Inset: enlarged kinetics of electrons in first 40 ps.

As shown in Figure R5, in first 40 ps there is no difference between the kinetics in CO₂ and argon because the main step in this relatively short temporal domain is the charge carrier generation and

trapping (Chem. Rev. 1995, 95, 69). After these steps, the trapped electrons were transferred to the absorbed target molecules in the temporal domain of 50-500 ps. Compare to the inert gas molecule of argon, the CO₂ molecule could accept the photogenerated electron thus showing a decrease in the temporal domain of 50-500 ps. This phenomenon is consistent with the results shown by TR-PL (Figure 4c).

We have revised the description in the Manuscript as follows.

“For both N-CP-D (Fig. 4c) and L-CP-D (inset of Fig. 4c), the average lifetime of electrons (Supplementary Table 8) in a CO₂ atmosphere was significantly shortened compared to that in an argon atmosphere, which can be mainly attributed to the large amount of photogenerated electrons can be fast delivered to CO₂ via the cocatalyst and only a small number of electrons with short lifetime involved into the detectable recombination process under the CO₂ atmosphere (Supplementary Fig. 34), which can be further confirmed by the TA spectra in different atmosphere (Supplementary Fig. 35).”

A few other more minor comments and questions:

1) At first, I was very excited about these results only to see later that TEOA was used. This is ok at this stage, but the manuscript needs to state clearer in the main-text that a scavenger was used.

Response: We have added the description of TEOA as the scavenger in the main text as follows.

“To study if the electron delivery from the CPs to cocatalyst has critical effects on CO₂ photoreduction properties, the evaluation of CO₂ photoreduction activities (see Methods for experimental details) were carried out in a closed gas circulation system by using CPs as the catalyst and 5 μmol Co (II) bipyridine complexes as cocatalyst. The acetonitrile/water (7:3) mixture with triethanolamine (TEOA) as sacrificial agent was also added.”

2) Net-like materials are usually referred to as conjugated microporous polymers in the materials community.

Response: We have added the description of the Net-like materials which are usually referred to as conjugated microporous polymers when we first mentioned Net-like materials in this revision.

“To validate the above strategy, four goal-oriented materials including linear and net-like (Net-like materials are usually referred to as conjugated microporous polymers) CPs with simple

structure^{24,25}, but different π -conjugations were built by using Suzuki-Miyaura coupling instead of Sonogashira-Hagihara coupling in synthesis.”

[24] Cooper, A. I. Conjugated microporous polymers. *Adv. Mater.* **21**, 1291-1295 (2009).

[25] Xu, Y., Jin, S., Xu, H., Nagai, A. & Jiang, D. Conjugated microporous polymers: design, synthesis and application. *Chem. Soc. Rev.* **42**, 8012-8031 (2013).

3) Comparing rates to other reports makes little sense as this ignores the fact that set-ups are very different and light sources vary (statement: ‘highest CO evolution efficiency’ and in a few other places). It would be much better to compare AQYs, which are already in the manuscript, to other reports.

Response: We thank the reviewer for this comment. We have added the comparison of AQYs with other similar CO₂ reduction systems in Table R4.

Table R4. Comparison of CO₂ photoreduction performance of the reported catalysts in the system similar to this work.

Catalyst	Cocatalyst	Sacrificial agent	Major products evolution rate	Enhancement after modification*	Selectivity	AQY	Ref.
L-CP-D	Co(bpy)₃Cl₂	TEOA	CO: 806 $\mu\text{mol g}^{-1}\text{h}^{-1}$	81-times	86.0 %	3.39 %	This work
N-CP-D	Co(bpy)₃Cl₂	TEOA	CO: 2274 $\mu\text{mol g}^{-1}\text{h}^{-1}$	138-times	82.0 %	1.23 %	This work
ZnIn ₂ S ₄ -In ₂ O ₃	Co(bpy) ₃ Cl ₂	TEOA	CO: 3075 $\mu\text{mol g}^{-1}\text{h}^{-1}$	3.5-times	~ 79.4 %	/	14
In ₂ S ₃ -CdIn ₂ S ₄	Co(bpy) ₃ Cl ₂	TEOA	CO: 825 $\mu\text{mol g}^{-1}\text{h}^{-1}$	12.0-times	~ 73.3 %	/	15
CPs-BT	Co(bpy) ₃ Cl ₂	TEOA	CO: 1213 $\mu\text{mol g}^{-1}\text{h}^{-1}$	4.6-times	81.6 %	1.75 %	16
CdS-BCN	Co(bpy) ₃ Cl ₂	TEOA	CO: 250 $\mu\text{mol g}^{-1}\text{h}^{-1}$	10.3-times	81.1 %	/	17
BCN	Co(bpy) ₃ Cl ₂	TEOA	CO: 94 $\mu\text{mol g}^{-1}\text{h}^{-1}$	/	76.2 %	/	18
HR-CN	Co(bpy) ₃ Cl ₂	TEOA	CO: 297 $\mu\text{mol g}^{-1}\text{h}^{-1}$	22.3-times	96.7 %	/	19
MCN/CoO _x	Co(bpy) ₃ Cl ₂	TEOA	CO: 204 $\mu\text{mol g}^{-1}\text{h}^{-1}$	2.76-times	78.5 %	0.25 %	20
CNU-BA	Co(bpy) ₃ Cl ₂	TEOA	CO: 1036 $\mu\text{mol g}^{-1}\text{h}^{-1}$	15.0-times	81.8 %	/	21
MOF-525-Co	/	TEOA	CO: 201 $\mu\text{mol g}^{-1}\text{h}^{-1}$	3.13-times	/	/	22

* The enhancement after modification were mentioned here due to the different researchers employed various evaluate system to measure the evolution rate.

We also mentioned in the Manuscript as follows.

“With the addition of appropriate cocatalyst (Supplementary Table 6), the CO selectivities of L-CP-D and N-CP-D was measured to be 86% and 82% (Supplementary Fig. 23) and achieved an apparent quantum yield (AQY) as high as 3.39 % and 1.23 % at 400 nm, respectively

(Supplementary Fig. 24), which is considered higher than that in most reports until now (Supplementary Table 4).”

4) Particle size and dispersion in the photocatalysis mixture are currently not considered. Some reports indicate that this might be important and should be measured and added.

Response: We thank the reviewer for this comment. From the TEM and SEM images, the particle sizes are different, and the net-like CPs are larger than the linear CPs. However, as shown in Figure R6, the dispersion of them in the solvent of acetonitrile/water (7:3) mixture is similar without adding Co complexes. After adding Co complexes, the CPs-A series still maintain the dispersion, while the CPs-D series stack together due to the intermolecular π - π stacking with the Co complexes. Although there are differences in dispersion among CPs, there are still good contacts between the materials and the solvent in the stirring condition. We have added this information to the Supplementary Information.

Figure R6. The dispersion of CPs in in the solvent of acetonitrile/water (7:3) mixture with or without adding Co complexes at different times.

5) Important experimental details are missing: What was the pressure of the photolysis experiment? This needs to be added to figure captions and also mentioned in the main-text. How much of the Co complex was used? I was a little unsure at times and it needs to be stated clearer.

Response: We thank the reviewer for this comment. The pressure of the photocatalytic experiments is about 80 kPa, and we have compared the different amounts (0, 1, 2, 5, 10 μmol) of the Co (II) bipyridine complexes as cocatalyst in Figure S20 of Supplementary Information as it is critical for the photocatalytic efficiency. In the main text, we compare the activities by adding 5 μmol of the Co (II) bipyridine complexes. We have added these both in the figure captions and the main text in this revision.

In the Figure caption:

“Fig. 3. CO₂ photoreduction performance of the CPs. a, Time course of the produced CO for CPs during 5 h experiment performed under visible light (420 nm cut-off filter) in an acetonitrile/water (7:3) mixture using triethanolamine (TEOA) as sacrificial agent and 5 μmol Co (II) bipyridine complexes as a cocatalyst over 5 mg CPs under \sim 80 kPa of pure CO₂ gas.”

In the Manuscript:

“To study if the electron delivery from the CPs to cocatalyst has critical effects on CO₂ photoreduction properties, the evaluation of CO₂ photoreduction activities (see Methods for experimental details) were carried out in a closed gas circulation system by using CPs as the catalyst and 5 μmol Co (II) bipyridine complexes as cocatalyst. The acetonitrile/water (7:3) mixture with triethanolamine (TEOA) as sacrificial agent was also added.”

6) CO₂ absorption of a dry powder can be very different compared to a measurement under wet conditions. This has been reported for conjugated microporous polymers and should be measured if it is considered to be important.

Response: We thank the reviewer for this comment. The CO₂ absorption of a dry powder is very different compared to measurement under wet conditions based on the previous report (J. Am. Chem. Soc. 2012, 134, 10741). The chemisorption of CO₂ is a more important factor than CO₂ adsorption in CO₂ photoreduction. We therefore employed the in-situ infrared technology to investigate the adsorption and chemisorption of CO₂ over CPs in a solvent-containing environment (Figure R4).

Figure R4. In-situ FT-IR spectra of N-CP-D with Co complexes for the adsorption of CO₂ (a) and the chemisorption of CO₂ (b) in a solvent-containing environment.

As shown in Figure R4, although the CP-A series exhibited greater CO₂ adsorption than the CPs-D series, the L-CP-D and N-CP-D showed an enhanced intensity both in the region of CO₂ adsorption (a) and the region of CO₂ chemisorption (b) than L-CP-A and N-CP-A. It suggests that the CO₂ adsorption capacity of CPs-D series is stronger than that of CPs-A series under the solvent-containing environment, which provides favorable conditions for the subsequent CO₂ reduction reaction.

We have revised the following sentence in Manuscript for the explanation of the CO₂ adsorption.

“Nevertheless, the CO₂ absorption of a dry powder is very different from the wet conditions⁵⁰, the in-situ FT-IR indicates L-CP-D and N-CP-D exhibit enhancement both in CO₂ adsorption and CO₂ chemisorption than L-CP-A and N-CP-A (Supplementary Fig. 30). It means that the CO₂ adsorption capacity of CPs-D series is stronger than that of CPs-A series under the solvent-containing environment, which provides favorable conditions for the subsequent CO₂ reduction reaction.”

⁵⁰ Dawson, R. *et al.* Impact of water coadsorption for carbon dioxide capture in microporous polymer sorbents. *J. Am. Chem. Soc.* **134**, 10741-10744 (2019).

7) Selectivity is only briefly discussed in the main-text but omitted from Fig.3. I am a little unsure how important it is right now as the area is in its infancy, but it seems to be standard practise in the field and should be added.

Response: We have discussed the selectivity in the Supplementary Information based on the Figure S20. Based on the report with a similar system the obtained selectivity from the CPs-D series is not the highest one because the CPs could generate H₂. However, after adding with Co complexes, the photogenerated electrons were oriented transferred to the cocatalyst and the selectivity of CO increased with the amount of cocatalyst and eventually reached 82% and 86% for N-CP-D and L-CP-D, respectively, which is considered to be above the current level of reports. In addition, we also added the comparison of the selectivity of similar system for CO₂ reduction in Table S4 and revised the description in the Manuscript as follows.

“With the addition of appropriate cocatalyst (Supplementary Table 6), the CO selectivities of L-CP-D and N-CP-D was measured to be 86% and 82% (Supplementary Fig. 23) and achieved an apparent quantum yield (AQY) as high as 3.39 % and 1.23 % at 400 nm, respectively (Supplementary Fig. 24), which is considered higher than that in most reports until now (Supplementary Table 4).”

8) Were the TR-PL measurements in Fig.4 performed in the presence of a scavenger?

Response: We thank the reviewer for this comment. We have added the experimental details of TR-PL in the Supplementary Information, which also list as follows.

“**TR-PL measurement.** Photoluminescence spectra decay curves were obtained by using a Hamamatsu instrument (Hamamatsu, Japan) with a 1kHz Ti: sapphire regenerative amplifier (Solstice, Spectra-Physics) was used as an excitation light source of an optical parametric amplifier (OPA) (TOPAS prime, NIR-UV-Vis. LIGHT CONVERSION Inc.). The CPs were dispersed in the solvent of acetonitrile and water solution with scavenger of triethanolamine. The argon was firstly pumped into a sealed cell for 40 min by using a pipe inlet to exclude the dissolved oxygen and other atmosphere. After the above pre-treatment process, the TR-PL signals were recorded and these samples are named L-CP-A in Ar, L-CP-D in Ar, N-CP-A in Ar and N-CP-D in Ar, respectively, according to the added catalyst. As comparison, the CO₂ was subsequently pumped into a sealed cell for 40 min by using a pipe inlet. The TR-PL signals were also recorded and these samples are named L-CP-A in CO₂, L-CP-D in CO₂, N-CP-A in CO₂ and N-CP-D in CO₂, respectively.”

The TR-PL measurements in Figure 4 were performed in the presence of scavengers. In fact, these

TR-PL measurements are in situ measurements to obtain the real information of electron transfer between CPs and cocatalyst.

Reviewer #3 (Remarks to the Author):

The manuscript of J.H. Ye et al. reports on the original construction of intermolecular cascaded π -conjugation channels for powering CO₂ photoreduction by modifying both intramolecular and intermolecular conjugation of conjugated polymers. This study is of great interest and reports a rather novel approach. Detailed experimental analysis and theoretical calculations were conducted to verify the proposed approach and conclusion. This work might interest those in pursuit of better photogenerated electron delivery from the viewpoint of the molecular architecture. Though I inclined to recommend the publication with Nat. Commun., there are also some important revisions need to be made:

The authors thank the reviewer for the valuable comments. These comments are very helpful for improving the quality and value of this article.

1. Could you please give the loading content of the Co (II) bipyridine complexes cocatalyst on the CPs as this may be critical for the photocatalytic efficiency?

Response: We thank the reviewer for this comment. We also believe that the content of the Co (II) bipyridine complexes cocatalyst on the CPs is critical for the photocatalytic efficiency. As mentioned in Supplementary information, the production of H₂ decreased with an increasing amount of cocatalyst, while the production of CO gradually increased. The increase in CO production over L-CP-D was not apparent after adding more than 1 μ mol of cocatalyst, and this phenomenon appeared in the N-CP-D system after adding more than 5 μ mol of cocatalyst. With the addition of 10 μ mol of cocatalyst, the CO evolution selectivity of L-CP-D and N-CP-D obtained through our experiment reached 86% and 82%, respectively.

In addition, we also employed ICP-AES measurement to determine the loading content of the Co species on the CPs (Table R5) when adding with 5 μ mol of Co (II) bipyridine complexes cocatalyst. The loading content of the Co species on the CPs is due to the difference in the specific

surface area. Moreover, the CO₂ photoreduction activity over N-CP-D (11.37 μmol h⁻¹) is much higher than that over N-CP-A (0.08 μmol h⁻¹). Despite the different loading amount of Co (II) bipyridine complexes on N-CP-D and N-CP-A, the enhancement (~138 folds) of activities over N-CP-D is much higher than the enhanced loading amount (~2.33 folds) of Co complexes. It demonstrates that the main reason of enhanced activities is attribute to the intermolecular conjugate interactions between N-CP-D and Co complexes.

Table R5. The loading content of Co species detected by the inductively coupled plasma atomic emission spectroscopy (ICP-AES).

	L-CP-D	N-CP-D	L-CP-A	N-CP-A
Co (wt.%)	0.862	3.614	0.526	1.548

We have added the following sentences for the explanation of the loading content of Co species in Supplementary information.

In addition, the ICP-AES measurement was employed to determine the loading amount of the Co species on the CPs (Table S6) when adding with 5 μmol of Co (II) bipyridine complexes cocatalyst. The main reason of improved activities over CPs-D series is attribute to the enhanced intermolecular conjugate interactions.

2. As the author showed in Table S4, there are some residual palladium and copper metals detected in the CPs. Please explain if these metals, especially the Pd, will have any activity in the photocatalytic process.

Response: Indeed, there are trace amounts of residual palladium retains in all CPs. Despite it is inevitable that trace amounts of palladium remained in the CPs, all CPs without adding Co complexes as cocatalyst showed negligible photocatalytic CO₂ reduction activity. After adding cocatalyst, the CP-A series still exhibited low activities in CO₂ reduction, while the CP-D series showed a great enhancement, revealing the enhanced activities are attributed to the intermodular electron transfer between CP-D series and Co complexes.

We have added the following sentences for the explanation that the residual Pd has negligible effect on the photocatalytic CO₂ reduction activity in Manuscript.

All CPs showed quite low activity in the absence of cocatalyst, as well as the CP-D series in the presence of isolated cobalt chloride or dipyriddy (Supplementary Fig. S22), implying that the residual Pd has negligible effect on the photocatalytic CO₂ reduction activity.

3. In this study, the LUMO and HOMO were obtained by hybrid density-functional-theory-based first-principles calculations. The experimental verifications for these values by UPS or Mott-Schottky plot should be conducted.

Response: Based on previous reports, the energy band structures of these CPs were investigated via cyclic voltammetry measurement, which was regarded as the one of most suitable way to determine the LUMO and HOMO levels of these conjugated polymers (Adv. Mater. 2015, 27, 6265; Angew. Chem. Int. Ed. 2015, 54, 13594; Angew. Chem. Int. Ed. 2016, 55, 9783; Angew. Chem. Int. Ed. 2016, 55, 9202;). Combining with the bandgap energy from diffuse reflectance spectrum (DRS), the energy levels of these CPs were precisely positioned and showed in the Supplementary Table S1. According to the reviewer's suggestion, we also conducted the Mott-Schottky test (Figure R1) and UPS spectra (Figure R2). to further explore the band structures of these CPs that are coincidence well with our cyclic voltammetry (CV) results..

Figure R1. Mott-Schottky plots for CPs in 0.1 M Na₂SO₄ at pH=7 by using Ag/AgCl with saturated KCl as reference electrode.

As shown in Figure R1. The flat-bands of L-CP-D, L-CP-A, N-CP-D and N-CP-A were estimated to be -1.35, -0.76, -1.22, -0.65 V versus the Ag/AgCl (-1.15, -0.56, -1.02, -0.45 V versus normal hydrogen electrode), respectively. The flat-band potential usually exhibits a potential difference of 0-0.1 V to the LUMO levels (Nat. Mater. 2019, 18, 827), the LUMO of L-CP-D, L-CP-A, N-CP-D and N-CP-A were determined to be -1.25, -0.66, -1.12, -0.55 V versus normal hydrogen electrode, respectively. With the assistant of band gap from DRS measurement, the energy levels of these CPs were showed in Table R1, which showed high accordance with the energy levels determined by cyclic voltammetry measurement.

Table R1. The energy levels within the L-CP-D, L-CP-A, N-CP-D and N-CP-A obtained from the Mott-Schottky test.

	L-CP-D	N-CP-D	L-CP-A	N-CP-A
Band gap	2.62 eV	2.47 eV	2.10 eV	1.98 eV
HOMO	1.37 V	1.35 V	1.44 V	1.43 V
LUMO	-1.25 V	-1.12 V	-0.66 V	-0.55 V

Figure R2. UPS spectra in the cutoff and the onset energy regions of L-CP-D, L-CP-A, N-CP-D and N-CP-A.

As shown in Figure R2, ultraviolet photoelectron spectroscopy (UPS) was employed to evaluate the energy levels of CPs. The cutoff (E_{cutoff}) and onset (E_i) energy regions are shown in the UPS spectra, respectively. According to the equation $\phi = 21.2 - (E_{\text{cutoff}} - E_i)$, the HOMO levels of the L-CP-D, L-CP-A, N-CP-D and N-CP-A were calculated to be 1.34, 1.23, 1.46, 1.45 V versus normal hydrogen electrode, respectively. With the assistant of band gap analysis from DRS measurement, the energy levels of CPs were showed in Table R2, which showed high accordance with the energy levels determined by cyclic voltammetry measurement and Mott-Schottky test.

Table R2. The energy levels within the L-CP-D, L-CP-A, N-CP-D and N-CP-A obtained from the UPS test.

	L-CP-D	N-CP-D	L-CP-A	N-CP-A
Band gap	2.62 eV	2.47 eV	2.10 eV	1.98 eV
HOMO	1.34 V	1.23 V	1.46 V	1.45 V
LUMO	-1.28 V	-1.24 V	-0.64 V	-0.53 V

In summary, we have revised the discussion of the band structure in the Manuscript as follow:

“Cyclic voltammetry (CV) measurements were also conducted, the HOMO position can be determined by the irreversibility of the oxidation peaks due to the irreversible oxidation process of the CPs at the impressed voltage (Supplementary Fig. 10) revealed different energy levels within the CPs (Supplementary Table 1)³⁹. In addition, their energy levels were further investigated by the UPS (Ultraviolet Photoelectron Spectroscopy) (Supplementary Fig. 11) and Mott-Schottky test (Supplementary Fig. 12), which showed a high accordance with the energy levels determined by CV measurement (Supplementary Table 2).”

4. Authors mentioned that “The conductivity transients and calculated charge mobilities for CPs are displayed in Fig. 2a, in which the L-CP-A, with a linear structure and alkynyl group, exhibits charge mobility (μ_{tot}) of $0.32 \text{ cm}^2 \text{ V}^{-1} \text{ s}^{-1}$ ($\phi \Sigma\mu = 7.4 \times 10^{-5} \text{ V}^{-1} \text{ s}^{-1}$). As expected, the charge mobility of L-CP-D in the absence of alkynyl decreased to a much lower value of $0.15 \text{ cm}^2 \text{ V}^{-1} \text{ s}^{-1}$ ($\phi \Sigma\mu = 3.4 \times 10^{-5} \text{ V}^{-1} \text{ s}^{-1}$).” (Page 6). However, there is no corresponding results shown in Fig.2a. Please check this carefully.

Response: We are sorry for this careless mistake. We have mistaken the graphic symbol of the inset in Figure 2a. Actually, the graphic symbol in the inset in Figure 2a is L-CP-D and L-CP-A. We have revised it in this revision.

Figure R3. Conductivity transients observed by flash-photolysis time-resolved microwave spectroscopy upon excitation at 355 nm laser pulses at 9.1×10^{15} photons cm^{-2} for CPs.

5. Authors claimed that “Besides, the net-like CPs (N-CP-A and N-CP-D) possess more cocatalyst absorption sites of phenyl in the units, thus resulting in enhanced performance of net-like CPs (N-CP-A and N-CP-D) than those of linear CPs (L-CP-A and L-CP-D)” (Page 9). As Fig.4a shows, the N-CPs exhibited greater CO_2 adsorption than the L-CPs. Does the CO_2 adsorption have any effects on the CO_2 photoreduction efficiency except for the cocatalyst absorption sites number? This may be a result of multi-factors.

Response: We thank the reviewer for this comment. The CO_2 absorption of a dry powder is very different compared to measurement under wet conditions based on the previous report (J. Am. Chem. Soc. 2012, 134, 10741). The chemisorption of CO_2 is a more important factor than CO_2 adsorption in CO_2 photoreduction. We therefore employed the in-situ infrared technology to investigate the adsorption and chemisorption of CO_2 over CPs in a solvent-containing environment (Figure R4).

Figure R4. In-situ FT-IR spectra of N-CP-D with Co complexes for the adsorption of CO₂ (a) and the chemisorption of CO₂ (b) in a solvent-containing environment.

As shown in Figure R4, although the CP-A series exhibited greater CO₂ adsorption than the CPs-D series, the L-CP-D and N-CP-D showed an enhanced intensity both in the region of CO₂ adsorption (a) and the region of CO₂ chemisorption (b) than L-CP-A and N-CP-A. It suggests that the CO₂ adsorption capacity of CPs-D series is stronger than that of CPs-A series under the solvent-containing environment, which provides favorable conditions for the subsequent CO₂ reduction reaction.

We have revised the following sentence in Manuscript for the explanation of the CO₂ adsorption.

“Nevertheless, the CO₂ absorption of a dry powder is very different from the wet conditions⁵⁰, the in-situ FT-IR indicates L-CP-D and N-CP-D exhibit enhancement both in CO₂ adsorption and CO₂ chemisorption than L-CP-A and N-CP-A (Supplementary Fig. 30). It means that the CO₂ adsorption capacity of CPs-D series is stronger than that of CPs-A series under the solvent-containing environment, which provides favorable conditions for the subsequent CO₂ reduction reaction.”

⁵⁰ Dawson, R. *et al.* Impact of water coadsorption for carbon dioxide capture in microporous polymer sorbents. *J. Am. Chem. Soc.* **134**, 10741-10744 (2019).

6. There is a clerical error in SI (page 46). “As shown in Figure S4” should be “As shown in Figure S34”.

Response: We are sorry for this clerical error. We have corrected it in this revision and also checked the whole Manuscript and Supplementary Information to avoid the other clerical errors.

Then, I would highly suggest these minor revisions to be thoroughly addressed before re-submission.

We carefully read your comments and made corresponding modifications to each of them.

REVIEWERS' COMMENTS:

Reviewer #1 (Remarks to the Author):

The authors have addressed all the comments. It is ready for publication.

Reviewer #2 (Remarks to the Author):

The revisions have improved the manuscript significantly and I would recommend for it to be accepted for publication.

Reviewer #3 (Remarks to the Author):

The manuscript of J.H. Ye et al. reports on the original construction of intermolecular cascaded π -conjugation channels for powering CO₂ photoreduction by modifying both intramolecular and intermolecular conjugation of conjugated polymers. This study is of great interest and reports a rather novel approach. Detailed experimental analysis and theoretical calculations were conducted to verify the proposed approach and conclusion. This work might interest those in pursuit of better photogenerated electron delivery from the viewpoint of the molecular architecture. The conclusions are original and the work is convincing. Sufficient experiments are conducted to verify the conclusions. I think this work is now publishable after the modifications.

Response to Reviewers' Comments

REVIEWERS' COMMENTS:

Reviewer #1 (Remarks to the Author):

The authors have addressed all the comments. It is ready for publication.

Response: We are grateful to this referee for the efforts have been made on reviewing this manuscript and the valuable comments for improving the quality of this manuscript.

Reviewer #2 (Remarks to the Author):

The revisions have improved the manuscript significantly and I would recommend for it to be accepted for publication.

Response: We are grateful to this referee for the efforts have been made on reviewing this manuscript and the valuable comments for improving the quality of this manuscript.

Reviewer #3 (Remarks to the Author):

The manuscript of J.H. Ye et al. reports on the original construction of intermolecular cascaded π -conjugation channels for powering CO₂ photoreduction by modifying both intramolecular and intermolecular conjugation of conjugated polymers. This study is of great interest and reports a rather novel approach. Detailed experimental analysis and theoretical calculations were conducted to verify the proposed approach and conclusion. This work might interest those in pursuit of better photogenerated electron delivery from the viewpoint of the molecular architecture. The conclusions are original and the work is convincing. Sufficient experiments are conducted to verify the conclusions. I think this work is now publishable after the modifications.

Response: We are grateful to this referee for the efforts have been made on reviewing this manuscript and the valuable comments for improving the quality of this manuscript.